https://doi.org/10.1038/s42003-022-03066-9　　**OPEN**
# Topological and enzymatic analysis of human Alg2 mannosyltransferase reveals its role in lipid-linked oligosaccharide biosynthetic pathway

Meng-Hai Xiang[1,4], Xin-Xin Xu[1,4], Chun-Di Wang[1], Shuai Chen[1], Si Xu[1], Xiang-Yang Xu[2], Neta Dean[3], Ning Wang [1✉] & Xiao-Dong Gao [1✉]

N-glycosylation starts with the biosynthesis of lipid-linked oligosaccharide (LLO) on the endoplasmic reticulum (ER). Alg2 mannosyltransferase adds both the α1,3- and α1,6-mannose (Man) onto ManGlcNAc$_2$-pyrophosphate-dolichol (M$_1$Gn$_2$-PDol) in either order to generate the branched M$_3$Gn$_2$-PDol product. The well-studied yeast Alg2 interacts with ER membrane through four hydrophobic domains. Unexpectedly, we show that Alg2 structure has diverged between yeast and humans. Human Alg2 (hAlg2) associates with the ER via a single membrane-binding domain and is markedly more stable in vitro. These properties were exploited to develop a liquid chromatography-mass spectrometry quantitative kinetics assay for studying purified hAlg2. Under physiological conditions, hAlg2 prefers to transfer α1,3-Man onto M$_1$Gn$_2$ before adding the α1,6-Man. However, this bias is altered by an excess of GDP-Man donor or an increased level of M$_1$Gn$_2$ substrate, both of which trigger production of the M$_2$Gn$_2$(α-1,6)-PDol. These results suggest that Alg2 may regulate the LLO biosynthetic pathway by controlling accumulation of M$_2$Gn$_2$ (α-1,6) intermediate.

[1] Key Laboratory of Carbohydrate Chemistry and Biotechnology, Ministry of Education, School of Biotechnology, Jiangnan University, Wuxi, China. [2] Zaozhuang Jienuo enzyme co., Ltd, Zaozhuang, China. [3] Department of Biochemistry and Cell Biology, Stony Brook University, Stony Brook, NY, USA. [4] These authors contributed equally: Meng-Hai Xiang, Xin-Xin Xu. ✉email: wangning@jiangnan.edu.cn; xdgao@jiangnan.edu.cn

Asparagine (N)-linked glycosylation in eukaryotes is crucial for the function of glycoproteins. Although the diversity of N-linked glycan structures is immense, all these different glycans are derived from a common lipid-linked oligosaccharide (LLO) precursor. This $Glc_3Man_9GlcNAc_2$ precursor ($G_3M_9Gn_2$) is made by the ordered sequential addition of sugars to dolichol pyrophosphate (PDol) at the ER membrane[1,2] by twelve Alg (Asparagine-linked glycosylation) glycosyltransferases (GTases)[3,4]. The first seven sugars are added on the cytoplasmic side of the ER membrane to make $M_5Gn_2$-PDol intermediate, using nucleotide sugar donors uridine diphosphate N-acetylglucosamine (UDP-GlcNAc) and guanosine diphosphate mannose (GDP-Man)[5–7]. $M_5Gn_2$-PDol flips across the ER membrane, where the next seven sugars are added in the lumen using lipid-linked sugar donors. Once assembled, the oligosaccharide is transferred to nascent proteins by oligosaccharyltransferase. Defects in LLO synthesis or transfer to proteins result in severe phenotypes. In humans, congenital disorders of glycosylation (CDG) manifest as a broad range of multisystem disorders or embryonic lethality. Thus, a deeper understanding of LLO biosynthesis and regulation is important for understanding how defects in this process lead to disease.

A key intermediate in LLO synthesis is $M_5Gn_2$-PDol. The five mannoses on this oligosaccharide are added by the Alg1, Alg2 and Alg11 mannosyltransferases (MTases). Alg1 adds the first β1,4 mannose onto $Gn_2$-PDol to form $Man(β1,4)GlcNAc_2$-PDol ($M_1Gn_2$-PDol). Alg2 catalyzes addition of both the second and third mannose to $M_1Gn_2$-PDol to form a branched $Man(α1,3)Man(α1,6)$-$M_1Gn_2$-PDol ($M_3Gn_2$-PDol)[8]. Alg11 further elongates this branched product with two α1,2-linked mannoses to produce $M_5Gn_2$-PDol[9]. In addition, Alg1, Alg2 and Alg11 physically interact with one another and work as an MTase complex[7,10].

Among the different Alg GTases, Alg2 is particularly unusual in that it displays two distinct activities; the addition of a mannose in an α1,3 and in an α1,6 linkage. Evidence for these dual activities comes from both in vivo and in vitro experiments. In vivo, both $M_1Gn_2$- and $M_2Gn_2$-PDol intermediates accumulate in *alg2* mutants, including yeast[11,12], human[13] and the zygomycete *R. pusillus*[14,15]. In vitro, Alg2 immunoprecipitated from yeast and from human embryonic kidney cells (HEK293) displays both MTase activities[16,17]. Importantly, in a reconstituted system containing purified acceptor substrate, donor sugars, and neutral lipids, recombinant yeast Alg2 (ScAlg2) directly transfers an α1,3-Man, followed by an α1,6-Man onto a $M_1Gn_2$ to generate $M_3Gn_2$[18]. Quantitative MS analyses of these reconstituted reactions demonstrate that the α1,3- and α1,6-Man can be added to $M_1Gn_2$ in either order, suggesting the existence of alternative reaction routes for Alg2 to make $M_3Gn_2$[19]. A major hurdle in understanding the mechanism and potential regulatory circuits of these reactions comes from the inherent instability of ScAlg2, which may stem from its hydrophobicity. Indeed, optimal in vitro MTase assays of recombinant ScAlg2 require addition of crude membranes or neutral bilayer lipids[19]. This instability has thus far precluded a kinetic analysis of each of the individual Alg2 reaction steps.

Surprisingly, given the conservation of N-glycosylation, the yeast and predicted human Alg2 (hAlg2) structures have diverged considerably. Unlike ScAlg2, which has multiple membrane-binding domains (MBDs), hAlg2 lacks much of this hydrophobicity. In the present study, a topological and enzymatic analysis of hAlg2 protein was performed. We first determined that hAlg2 is anchored to the ER membrane by a single amphiphilic-like α helix. We found that recombinant hAlg2 purified from *E. coli* possesses markedly increased stability and higher MTase specific activity than ScAlg2. These properties were exploited to enable development of a sensitive in vitro hAlg2

MTase kinetic assay, in which each of the two reactions catalyzed by Alg2 could be distinguished and quantitated. Our results demonstrated that hAlg2 prefers to first add the α1,3-Man, followed by an α1,6-Man onto $M_1Gn_2$-pyrophosphate-phytanol (PPhy) to produce the branched $M_3Gn_2$-PPhy, but this bias can be altered by the concentration of both GDP-Man and substrate.

## Results

**Human Alg2 possesses a different ER membrane topology than its yeast counterpart.** A comparison of the predicted hydrophobicity of human and yeast Alg2 suggested these proteins diverged in their secondary structure. hAlg2 is predicted to possess a single hydrophobic region at its N-terminus, while ScAlg2 has four hydrophobic regions, including two helices that span the ER membrane at the N-terminus and two hydrophobic sequences at the C-terminus that bind the ER but do not traverse the membrane[17]. The ScAlg2 topology has been experimentally determined[16] but neither the ER localization nor the topology of hAlg2 has been studied.

We first confirmed the ER membrane localization of hAlg2 by immunofluorescence and by membrane fractionation. Immunofluorescence microscopy of FLAG-hAlg2 in HEK293 cells showed that it mostly co-localizes with the resident ER membrane protein Calnexin (CANX) (Fig. 1a). Fractionation of lysates from these cells by differential centrifugation was used to separate ER membrane and soluble proteins, as described in **Methods**. Western blot analysis of these fractions showed that FLAG-hAlg2 was mainly distributed in the ER membrane fraction (Fig. 1b). To determine if hAlg2 behaved as an integral membrane protein versus one that is peripherally associated with the ER, ER membrane fractions were treated with $Na_2CO_3$ (which releases peripheral but not integral membrane proteins) and 0.2% sodium dodecyl sulfate (SDS) detergent (which releases both peripheral and integral membrane proteins). Under these conditions, hAlg2 protein was released from the ER membrane by 0.2% SDS but not by $Na_2CO_3$ (pH 11.0) extraction (Fig. 1c), suggesting that hAlg2 associates with the ER as though it can span the membrane in HEK293 cells.

Based on the data described above, two models can explain hAlg2 membrane topology (Fig. 1d). Since Alg2 catalyzes a reaction in the cytosol and its conserved catalytic domain resides in the C-terminal domain, both models place the C-terminal domain in the cytosol. In the first model (Model 1), the N-terminal hydrophobic region works as a membrane-spanning transmembrane domain (TMD); in the second model (Model 2), the N-terminal hydrophobic region works as an MBD that interacts with, but does not cross the ER membrane.

**Human Alg2 protein attaches to the ER membrane with a single membrane-binding domain.** These two topological models for hAlg2 (Fig. 1d) differ by the orientation of its N-terminus. If hAlg2 has a single TMD, then the N-terminus is predicted to be in the lumen of the ER (Model 1). If hAlg2 interacts with the ER by a peripheral membrane binding domain, then the N-terminus is predicted to be in the cytosol (Model 2). To test these models, experiments were designed to map the orientation of the hAlg2 N-terminus.

The first set of experiments employed hAlg2 fused to a fragment of yeast invertase (Suc2A). The Suc2A fragment contains three NX(S/T) glycosylation sites, whose glycosylation acts as a readout for assessing if this fragment is in the ER lumen[20–23]. As depicted in Fig. 2a, the Suc2A cassette together with a FLAG tag were fused at either the N- and C-termini of hAlg2, resulting in FLAG-Suc2A-hAlg2 and hAlg2-Suc2A-FLAG (Table 1). The FLAG-Suc2A-hAlg2 was designed for mapping the

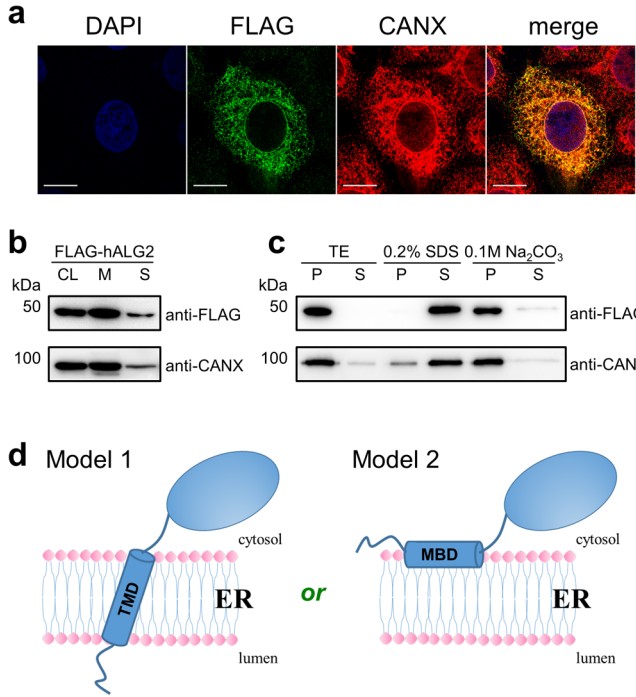

**Fig. 1 Subcellular localization of hAlg2 protein. a** Immunofluorescence of FLAG-tagged hAlg2 protein. HEK293 cells transiently expressing N-terminally FLAG-tagged hAlg2 were doubly immunostained for FLAG and ER marker CANX (Calnexin), followed by Alexa Fluor 488-conjugated anti-mouse and Alexa Fluor 555-conjugated anti-rabbit secondary antibodies. Images were obtained by confocal microscopy. Nuclei were stained with DAPI (blue). Green indicates FLAG-hAlg2. Red indicates CANX. Merge indicates co-localization. Bar, 20 μm. **b** Fractionation of FLAG-hAlg2 and detected by western blotting. CL represents cell lysate, M represents membrane proteins, S represents soluble proteins. **c** Membrane extraction of FLAG-hAlg2 protein. The membrane fraction was treated with 0.2% SDS or 0.1 M $Na_2CO_3$ (pH 11) and further centrifuged at 100,000 × $g$ to separate membrane and soluble proteins. Equivalent amounts of proteins from the pellet (P) and supernatant (S) were compared by western blotting. **d** Putative topological models of hAlg2 on the ER.

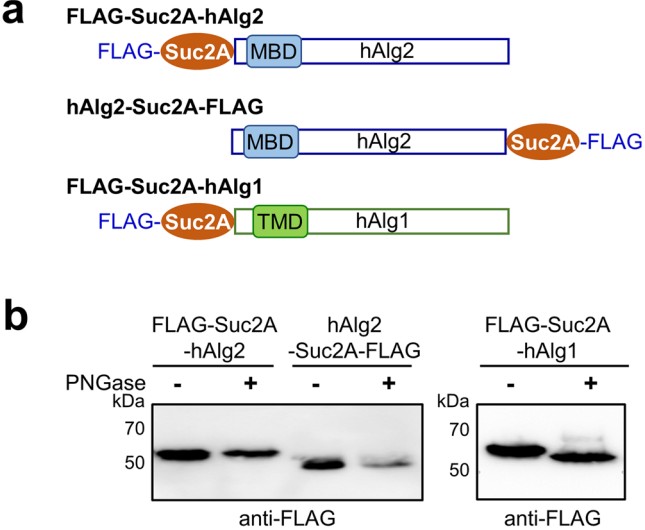

**Fig. 2 Mapping the orientation of hAlg2 N-terminus. a** Schematic representation of the FLAG-Suc2A-hAlg2, hAlg2-Suc2A-FLAG and FLAG-Suc2A-hAlg1 constructs. The hAlg2 was fused to a yeast invertase fragment (Suc2A) and tagged with FLAG epitope. The Suc2A contains three NX(S/T) sites, which acts as a reporter for N-glycosylation. **b** Western blotting of FLAG-Suc2A-hAlg2, hAlg2-Suc2A-FLAG, and FLAG-Suc2A-hAlg1 expressed in HEK293 cells with or without PNGase F treatment.

### Table 1 Plasmids used in this study.

| Plasmid | Description |
| --- | --- |
| pME | Mammalian cell expression vector |
| YEp352GAPII | URA3/2μ yeast shuttle vector containing TDH3 promoter |
| pME-FLAG-hALG2 | FLAG-hAlg2 expressed in pME |
| pME-GFP | Enhanced GFP expressed in pME |
| pME-GFP-hALG2 | GFP-hAlg2 expressed in pME |
| pME-hALG2-GFP | hAlg2-GFP expressed in pME |
| pME-mRFP-KDEL | mRFP-KDEL expressed in pME |
| YEp352GAPII-FLAG-SUC2A-hALG2 | FLAG-Suc2A-hAlg2 expressed in YEp352GAPII |
| YEp352GAPII-hALG2-SUC2A-FLAG | hAlg2-Suc2A-FLAG expressed in YEp352GAPII |

orientation of the hAlg2 N-terminus, while the latter one was for confirming the cytosolic orientation of its C-terminus. When expressed in vivo, these Suc2-fusion hAlg2 proteins appeared enzymatically active as both could complement a yeast *alg2* mutant (Supplementary Fig. 1). As an additional control, an N-terminally Suc2-fused human Alg1 protein (Fig. 2a) was analyzed in parallel because the same Suc2A-hAlg1 fusion protein was used successfully to demonstrate the luminal orientation and hence N-glycosylation of the ScAlg1 N-terminus[24].

To determine their orientation, FLAG-Suc2A-hAlg2, hAlg2-Suc2A-FLAG, and FLAG-Suc2A-hAlg1 (Table 1) were transiently expressed in HEK293 cells. Proteins were extracted, treated with or without PNGase to cleave N-linked glycans and subjected to immunoblot analysis. If either terminus was luminally oriented and glycosylated, then treatment with PNGase would result in a decreased molecular weight on sodium dodecyl sulfate-polyacrylamide gel electrophoresis (SDS-PAGE). The results of this experiment demonstrated the molecular weight of both hAlg2 fusion proteins was unchanged after PNGase F treatment; both fusion proteins migrated with the same mobility as untreated control samples (Fig. 2b, left panel). In contrast, untreated FLAG-Suc2A-hAlg1 migrated with a larger molecular weight than the PNGase treated sample, indicating it had undergone N-glycan modification (Fig. 2b, right panel). These

results suggested that unlike the N-terminus of hAlg1, neither the N- or C-termini of hAlg2 undergo Suc2A dependent glycosylation and imply that both termini reside in the cytosol in HEK293 cells.

To provide additional evidence for the idea that both ends of hAlg2 are in the cytosol, we analyzed its membrane topology using a protease protection assay, in which domain susceptibility to exogenously added protease is an indicator of cytosolic orientation. For this assay, hAlg2 was tagged with GFP at the N- or C-terminus (Fig. 3a) to enable a fluorescence protease protection (FPP) assay[25–27] (Fig. 3b). Since the plasma membrane of mammalian cells contains more cholesterol than intracellular membranes, it can be selectively permeabilized by certain detergents. Proteinase K treatment of these permeabilized cells leads to selective proteolysis and loss of fluorescence of a GFP tag that faces the cytosol (Fig. 3b, green heads), while a luminally-oriented GFP tag is inaccessible and thus protected from proteolysis (Fig. 3b, red heads). To perform this experiment, HEK293 cells expressing cytoplasmic GFP were first subjected to permeabilization with varying concentrations of digitonin for

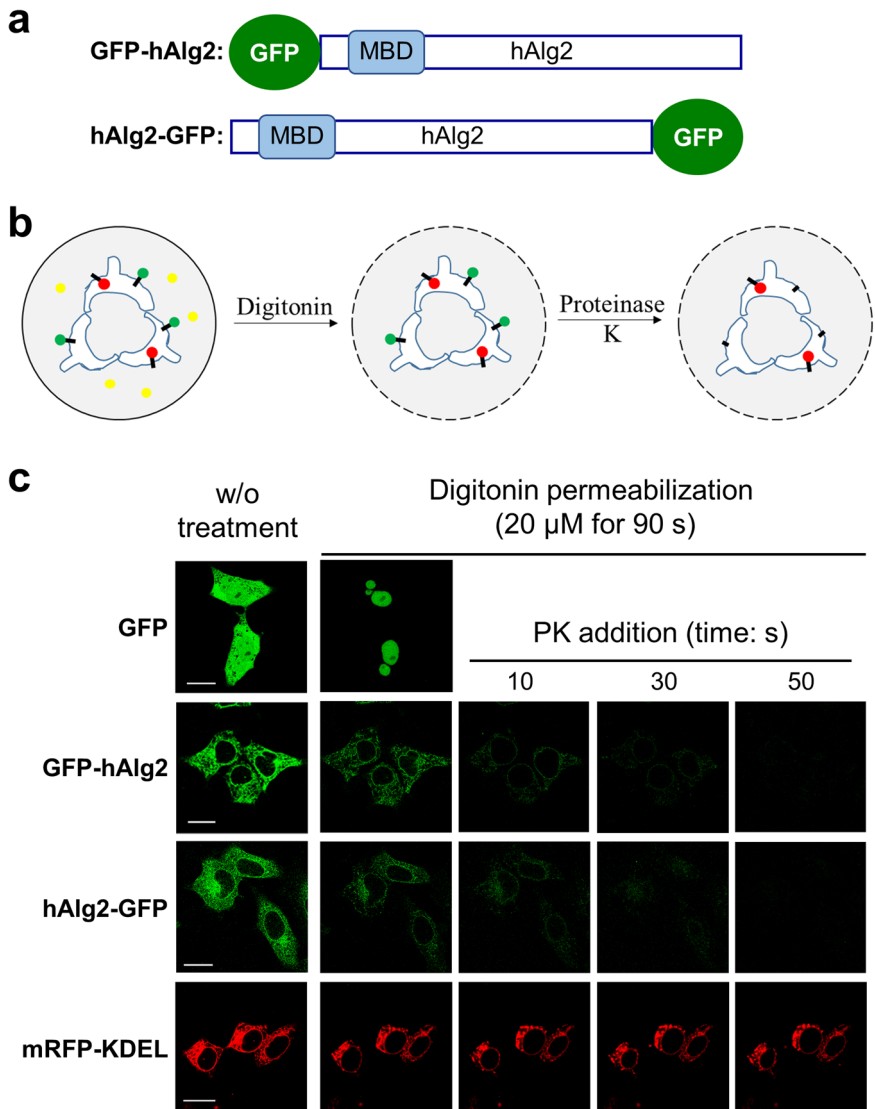

**Fig. 3 Confirmation of the cytosolic orientation of hAlg2 termini with fluorescence protease protection (FPP). a** Schematic representation of the GFP-hAlg2 and hAlg2-GFP constructs. **b** Mechanism of FPP assay. Green dots: GFP tags that face the cytosol; Red dots: RFP tags that face the ER lumen. **c** FPP assay of HEK293 cells express GFP, GFP-hAlg2, hAlg2-GFP or mRFP-KDEL. The digitonin treatment was carried out at a concentration of 20 μM for 90 s as permeabilization state for all experiments. Bar, 20 μm.

different times to optimize conditions, which were found to be 20 μM digitonin for 90 s (Supplementary Fig. 2).

Under these conditions, HEK293 cells transiently expressing GFP-hAlg2 or hAlg2-GFP were permeabilized and assayed by FPP. As controls, we also assayed HEK293 cells expressing mRFP-KDEL and GFP (Fig. 3c). mRFP-KDEL was used as an ER lumen marker[20–23] that should be protected from proteolysis by added protease, while GFP was used as a cytoplasmic marker that should be released from the cytosol upon permeabilization. Cells were imaged by confocal microscopy before permeabilization, post permeabilization but before adding protease, and post permeabilization after protease addition (Fig. 3c). After permeabilization but before adding protease, cells were apparently permeabilized since the cytoplasmic GFP fluorescence completely disappeared, presumably by diffusion. However, most of the signal from GFP-hAlg2 and hAlg2-GFP remained on the ER, indicating both fusion proteins were tightly bound to the ER. After addition of 50 μg/mL proteinase K, fluorescence of both GFP-hAlg2 and hAlg2-GFP were lost rapidly over time by 50 sec. These results indicated that the GFP tags placed on either end of

hAlg2 are on the cytoplasmic side of the ER. The mRFP-KDEL fluorescence was unaffected by both digitonin and proteinase K, even after prolonged incubation times (Fig. 3c), indicating that under these conditions digitonin did not permeabilize the ER membrane and ER luminal contents remained completely protected from proteolysis. Collectively, our results from two different assays provided evidence that both the N- and C-termini of hAlg2 are oriented toward cytoplasmic side of the ER (Fig. 1d, Model 2). These results underscore the markedly different membrane topology between yeast and human Alg2; ScAlg2 has four hydrophobic domains, two of which cross the membrane while human Alg2 has a single hydrophobic domain that does not cross the membrane.

**Recombinant hAlg2 and ScAlg2 have similar bifunctional MTase activities, but hAlg2 is more active.** The topological differences between ScAlg2 and hAlg2 raised the question of whether or not they have the same enzyme activities. Previous studies demonstrated that recombinant ScAlg2 can add both

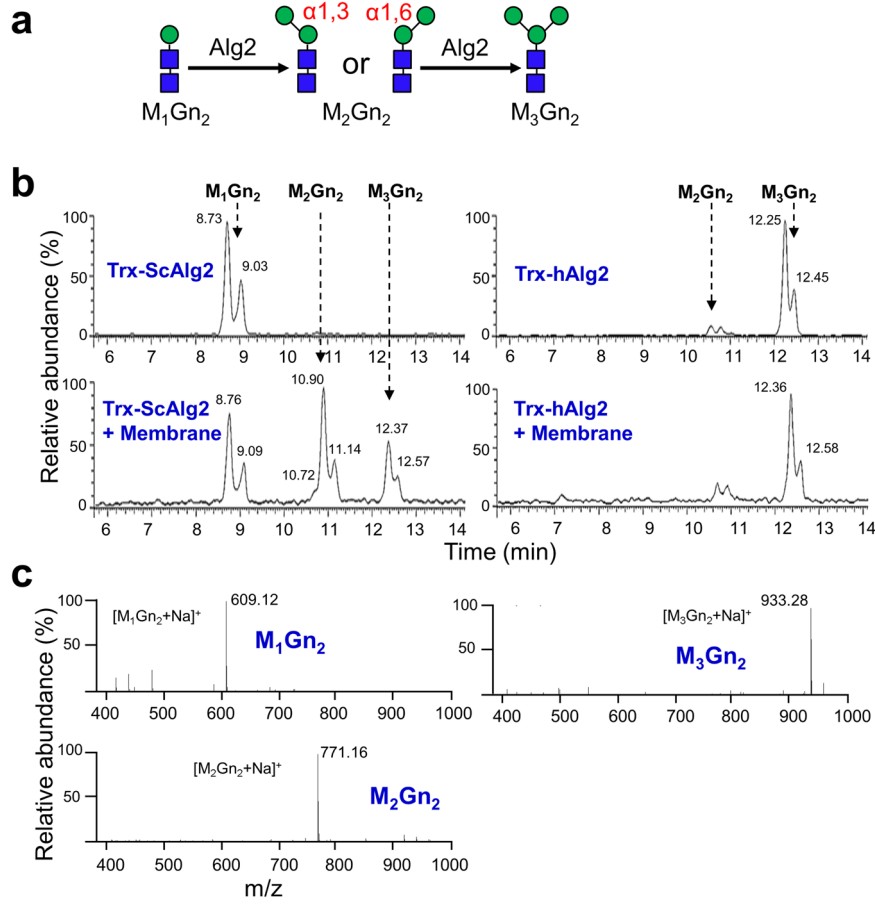

**Fig. 4 Comparison of MTase activity between hAlg2 and ScAlg2. a** Schematic diagram of the reaction catalyzed by Alg2. Alg2 transfers Man to $M_1Gn_2$ in either an α1,3- or α1,6-linkage, to produce two different $M_2Gn_2$ intermediates. **b** The UPLC chromatograms of glycan chains released from the Alg2 reaction mixtures. Reactions were performed using purified Trx-ScAlg2 or Trx-hAlg2 (40 ng/μL) with the standard reaction mixture. After mixing, heat-inactivated *E. coli* membrane was added upon request and reactions incubated at 30 °C for 4 h. **c** ESI-MS spectra of the glycans released from phytanyl oligosaccharides. Mass analyses showed the peaks eluted (Fig. 4b) at ~ 8.9 min, ~ 11.0 min and ~ 12.4 min correspond to $M_1Gn_2$ ($[M_1Gn_2 + Na]^+$), $M_2Gn_2$ ($[M_2Gn_2 + Na]^+$) and $M_3Gn_2$ ($[M_3Gn_2 + Na]^+$), respectively.

**Table 2 Relative percentage of oligosaccharides produced by Alg2 in various reaction conditions.**

| Purified protein | Crude membrane | $M_1Gn_2$ (%) | $M_2Gn_2$[a] (%) | $M_3Gn_2$ (%) |
|---|---|---|---|---|
| Trx-ScAlg2 | - | 100 | 0 | 0 |
| | + | 34 | 42 | 24 |
| Trx-hAlg2 | - | 0 | 14 | 86 |
| | + | 0 | 22 | 78 |

Relative percentage was calculated by dividing the area of each corresponding oligosaccharide peak by the total area of all peaks (×100) in the UPLC chromatograms are shown in Fig. 4b.
[a]$M_2Gn_2$ includes the $M_2Gn_2$ (α-1,3) and $M_2Gn_2$ (α-1,6) oligosaccharides.

α1,3-and α1,6-Man to the $M_1Gn_2$-PPhy in either order to form $M_3Gn_2$-PPhy (Fig. 4a)[19]. To determine if hAlg2 also catalyzes both of these reactions, we purified recombinant hAlg2 and ScAlg2 in parallel and compared their in vitro specific activities.

Recombinant ScAlg2 and hAlg2 proteins were expressed in *E. coli* and purified as described in **Methods**. SDS-PAGE analysis revealed a single band of about 68 kDa and 57 kDa, corresponding to thioredoxin-tagged ScAlg2 (Trx-ScAlg2) and Trx-hAlg2 respectively (Supplementary Fig. 3). To assay MTase activities, purified recombinant proteins (40 ng/μL) were added to a standard reaction, in which Man-(β1,4)-$Gn_2$-pyrophosphate-phytanyl ($M_1Gn_2$-PPhy) was used as the LLO acceptor instead

of the natural $M_1Gn_2$-PDol acceptor (see **Methods** for details). Phytanol, with only four isoprene units, is much shorter than dolichol ($C_{85-105}$) and can be easily synthesized[28]. Importantly, the first five MTases of lipid-linked oligosaccharide synthesis show no preference for dolichol- versus phytanol-linked substrates[3]. Therefore phytanol-linked oligosaccharides (PLOs) are widely used as acceptor substrates for enzymatic studies of Alg GTase[3,19,29,30]. After incubating for 4 h, glycan products including $M_1Gn_2$, $M_2Gn_2$, and $M_3Gn_2$ were chemically released from PPhy and analyzed by ultra-performance liquid chromatography-mass spectrometry (UPLC-MS) (Fig. 4b, c). Noteworthy, dual peaks observed in UPLC-MS are likely the result of α- and β-anomeric forms of released glycans. To compare their activities quantitatively, the percentage of each glycan product was calculated (Table 2). Although the $M_2Gn_2$ intermediate includes both α-1,3 and α-1,6 forms, their combined value was used for $M_2Gn_2$ calculation.

Our previous study demonstrated ScAlg2 in vitro MTase activity is dependent on the inclusion of membranes, either from *E. coli* or neutral lipids that can form bilayers[19]. Therefore, as expected, in the ScAlg2 reaction, glycan peaks corresponding to $M_2Gn_2$ intermediates or final $M_3Gn_2$ product were only observed if a membrane fraction prepared from *E. coli* (expressing empty vector) was added (Fig. 4b, left panel; Fig. 4c). After a 4 h reaction, 66% of the $M_1Gn_2$ substrate was converted to $M_2Gn_2$ (42%) and $M_3Gn_2$ (24%) by ScAlg2 (Table 2). In contrast,

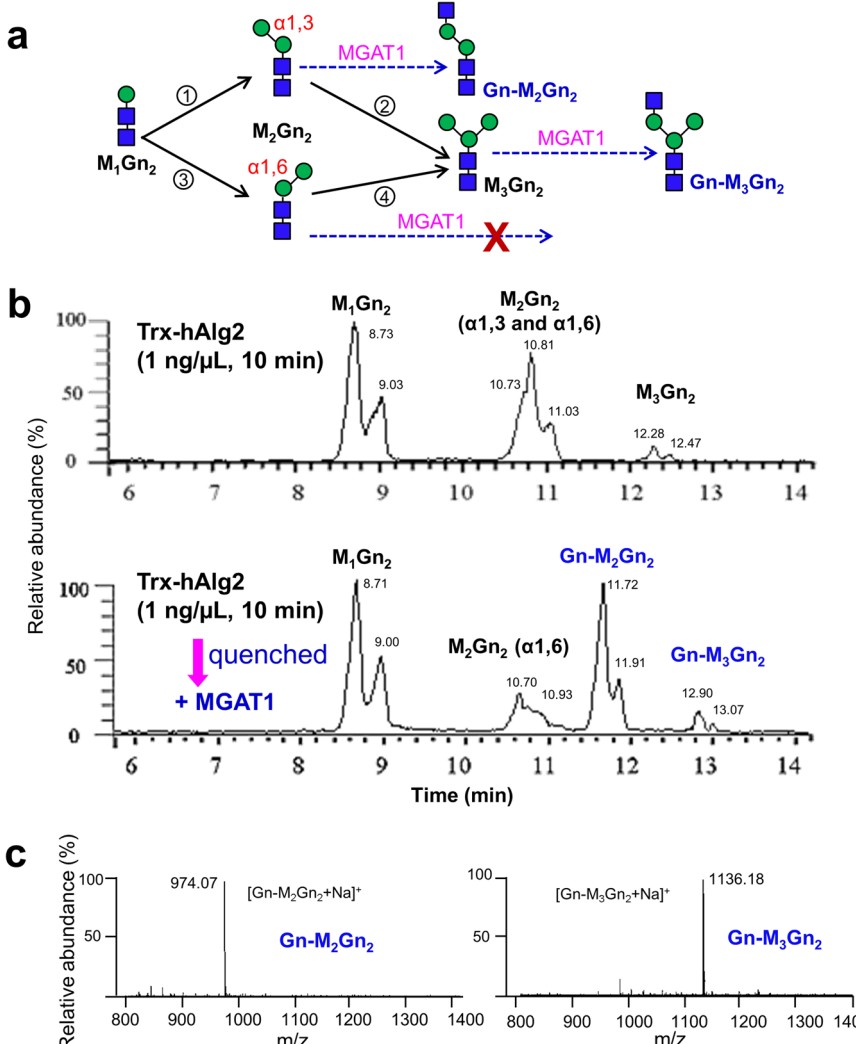

**Fig. 5 Separation of $M_2Gn_2$ intermediates by using MGAT1. a** Schematic diagram of the reactions catalyzed by Alg2 and MGAT1. MGAT1 recognizes the intermediate $M_2Gn_2$ (α-1,3) and product $M_3Gn_2$, and elongates these to produce $Gn-M_2Gn_2$ and $Gn-M_3Gn_2$, respectively. **b** Representative UPLC chromatograms of glycan chains released from the reaction mixtures. Top panel: the reaction included purified Trx-hAlg2 (1 ng/μL) with the standard reaction mixture for 10 min. Bottom panel: the reaction shown in Fig. 5b top panel was quenched by heating, followed by addition of MGAT1 and UDP-GlcNAc and further incubated for another 6 h. **c** ESI-MS spectra of the glycans released from phytanyl oligosaccharides. Mass analyses showed the peaks eluted (Fig. 5b, bottom panel) at ∼ 11.8 min and ∼ 13.0 min correspond to $Gn-M_2Gn_2$ ([$Gn-M_2Gn_2 + Na$]$^+$) and $Gn-M_3Gn_2$ ([$Gn-M_3Gn_2 + Na$]$^+$), respectively.

purified hAlg2 was capable of completely converting the substrate to intermediate or the final product $M_3Gn_2$ without added membrane (Fig. 4b, right panel; Fig. 4c). The substrate was completely converted by hAlg2 to products, yielding 14% of $M_2Gn_2$ and 86% of $M_3Gn_2$ (Table 2). Addition of membrane fraction to the hAlg2 reactions negatively skewed the in vitro activity, decreasing yield of $M_3Gn_2$ to 78% (Table 2). Thus, our results demonstrated that hAlg2 has a higher specific MTase activity that, unlike ScAlg2, is independent of added membrane in vitro.

**Establishing a quantitative assay for kinetic analyses of hAlg2 MTase activities**. The increased in vitro specific MTase activity, coupled with the ease with which hAlg2 can be assayed suggested the feasibility of developing a method to study the kinetics of each of the two Alg2 MTase activities that give rise to different $M_2Gn_2$ intermediates before forming $M_3Gn_2$. As a means of separating these two $M_2Gn_2$ intermediates, we took advantage of the

substrate specificity of human α-1,3-mannosyl-glycoprotein 2-beta-N-acetylglucosaminyltransferase 1 (MGAT1) (Fig. 5a). In vivo MGAT1 adds a GlcNAc residue to the C2 of the α1,3-Man residue in the core of $M_5Gn_2$ on mature glycoproteins[31–33]. Because of its rigid substrate specificity for α1,3-Man, we expected that MGAT1 would only recognize the $M_2Gn_2$ (α-1,3) intermediate and $M_3Gn_2$ product, and convert them to $Gn-M_2Gn_2$ (α-1,3) and $Gn-M_3Gn_2$ respectively while the $M_1Gn_2$ and intermediate $M_2Gn_2$ (α-1,6) would remain unmodified (Fig. 5a).

To test this idea, purified hAlg2 (1 ng/μL) was incubated in the standard reaction mixture for 10 min. $M_2Gn_2$ intermediates were detected as three overlapping peaks with a retention time of ∼ 11.0 min by UPLC (Fig. 5b, top panel), consistent with the production of both $M_2Gn_2$ (α-1,3) and $M_2Gn_2$ (α-1,6) isoforms. After quenching the reaction, purified MGAT1 was added and the reaction was incubated for another 6 h. In our previous study, we confirmed that under similar reaction conditions, purified recombinant MGAT1 completely converted peptide-modified $M_3Gn_2$ to $Gn-M_3Gn_2$ in 2 h[34]. Therefore, in our assay, a 6 h

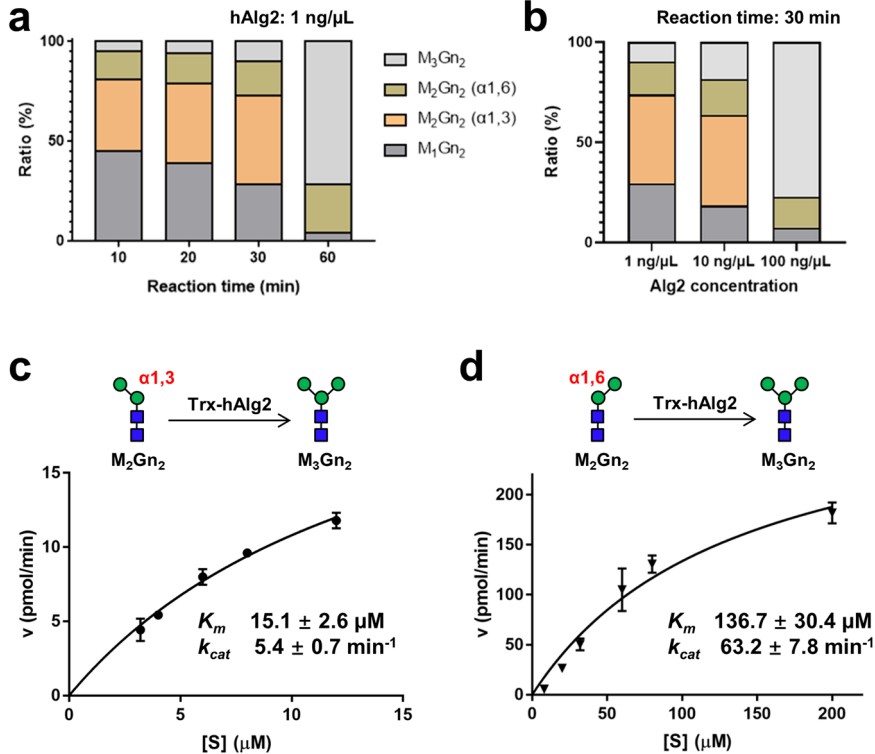

**Fig. 6 Kinetic analysis of M₃Gn₂ formation. a** The percentage of each component in the reaction as a function of time. The reaction was performed using purified Trx-hAlg2 (1 ng/µL) with the standard reaction mixture. **b** The percentage of each component in the reaction system as a function of Alg2 concentration. The reactions were performed using purified Trx-hAlg2 (1, 10 and 100 ng/µL) with the standard reaction mixture at 37 °C for 30 min. **c** The $K_m$ (15.1 ± 2.6 µM) and $k_{cat}$ (5.4 ± 0.7 min⁻¹) values for the substrate M₂Gn₂ (α-1,3) (black dots) were calculated by nonlinear regression (GraphPad Prism 7) with a constant concentration of 2 mM GDP-Man. Each data point represents the mean value calculated from n = 3 independent experiments. Error bars represent standard deviations. **d** The $K_m$ (136.7 ± 30.4 µM) and $k_{cat}$ (63.2 ± 7.8 min⁻¹) values for the substrate M₂Gn₂ (α-1,6) (black triangles) were calculated by nonlinear regression (GraphPad Prism 7) with a constant concentration of 2 mM GDP-Man. Each data point represents the mean value calculated from n = 3 independent experiments. Error bars represent standard deviations. Note the difference in values on the X and Y axes on graphs shown in **c**, **d**.

incubation was chosen for converting M₃Gn₂- or M₂Gn₂-PPhy intermediates. As shown in Fig. 5b (bottom panel), after treatment with MGAT1, a large portion of the M₂Gn₂ peaks was shifted to a retention time around 11.8 min, while a small amount remained at the retention time around 10.9 min. This shifted peak is consistent with the mass of [Gn-M₂Gn₂ + Na]⁺ (m/z: 974), the product of MGAT1 acting on M₂Gn₂ (α-1,3) (Fig. 5c), while the remaining peak represents the unmodified M₂Gn₂ (α-1,6) intermediate. In addition, the M₃Gn₂ product completely disappeared from the area with a retention time ~ 12.3 min, and shifted to a new position with a retention time of ~ 13.0 min with the mass peaks representing [Gn-M₃Gn₂ + Na]⁺ (m/z: 1136) (Fig. 5c). These results demonstrated the feasibility of this approach to kinetically dissect the dual activities of hAlg2 as this single reaction could simultaneously distinguish and quantitate all hAlg2 MTase intermediates and products.

**Human Alg2 prefers to add α1,3-Man onto M₁Gn₂-PPhy followed by the α1,6-Man.** Alg2 can catalyze four MTase reactions: the addition of an α1,3-Man onto M₁Gn₂-PDol and M₂Gn₂(α1,6)-PDol or the addition of an α1,6-Man onto M₁Gn₂-PDol and M₂Gn₂(α1,3)-PDol. These reactions can be divided into two M₃Gn₂ alternative synthesis routes (Fig. 5a, step 1 → 2 and step 3 → 4). We applied the in vitro quantitative assay described above to study the kinetics of forming M₃Gn₂ by each of these alternative synthetic routes as well as the parameters that bias one versus the other (Fig. 5).

Purified hAlg2 (1 ng/µL) was incubated in the standard reaction mixture. Samples were taken at the indicated incubation times and quenched. After the treatment with MGAT1, glycan products in the reactions were quantitatively analyzed by UPLC-MS. As shown in Fig. 6a, in the first 30 min, both α1,3- and α1,6-Man residues were attached to M₁Gn₂-PPhy but M₂Gn₂ (α-1,3) was produced at ~ two folds higher level than Mn2Gn₂ (α-1,6). As the reaction proceeded, M₂Gn₂ (α-1,3) was gradually converted to M₃Gn₂, and was completely converted by 60 min, while M₂Gn₂ (α-1,6) remained. This strong bias toward production of M₂Gn₂ (α-1,3) is similar to that seen with ScAlg2[19]. Incubation with an increased concentration of hAlg2 (from 1 ng/µL to 10 ng/µL) accelerated the rate with which M₁Gn₂-PPhy was converted to M₂Gn₂ and M₃Gn₂, and also led to a marked accumulation of M₂Gn₂ (α-1,3) as the major intermediate (Fig. 6b). Increasing the concentration of hAlg2 to 100 ng/µL further accelerated the reaction, which was almost completed in 30 min (Fig. 6b). Under this condition, only the M₂Gn₂ (α-1,6) intermediate was detected. These results suggested that M₂Gn₂ (α-1,3) is the preferred intermediate product of hAlg2, and also that M₂Gn₂ (α-1,3) is rapidly converted to the M₃Gn₂ by addition of second (α-1,6) linked Man residue.

To further analyze the preference of Alg2 for its secondary substrates, each of the two intermediates, M₂Gn₂ (α-1,3)- and M₂Gn₂ (α-1,6)-PPhy, were prepared as described in **Methods**, and used as substrates to measure the kinetics by which they are converted to M₃Gn₂-PPhy. Their rate of conversion to M₃Gn₂ by purified recombinant hAlg2 (5 ng/µL) was measured with a fixed

**Table 3 Relative percentage of oligosaccharides in hAlg2 reactions with 20 μM of $M_1Gn_2$-PPhy substrate.**

| GDP-Man (μM) | $M_1Gn_2$ (%) | $M_2Gn_2$ (%) (α-1,3/α-1,6) | $M_3Gn_2$ (%) |
|---|---|---|---|
| 5 | 75.1 | 24.1 (NA) | 0.8 |
| 10 | 72.7 | 25.4 (NA) | 1.9 |
| 25 | 62.8 | 34.8 (NA) | 2.4 |
| 200 | 36.7 | 40.2 (NA) | 23.1 |
| 2000 | 28.7 | 37.4 (17/1) | 33.9 |

Relative percentage was calculated by dividing the area of each corresponding oligosaccharide peak by the total area of all peaks (×100) in the UPLC chromatograms. The reactions were performed using purified Trx-hAlg2 (5 ng/μL) incubated with the indicated concentrations of GDP-Man for 30 min. NA, not available.

**Table 4 Relative percentage of oligosaccharides in hAlg2 reactions with 25 μM of GDP-Man donor.**

| $M_1Gn_2$-PPhy (μM) | $M_1Gn_2$ (%) | $M_2Gn_2$ (%) (α-1,3/α-1,6) | $M_3Gn_2$ (%) |
|---|---|---|---|
| 20 | 62.8 | 34.8 (NA) | 2.4 |
| 40 | 53.8 | 42.9 (38/1) | 3.3 |
| 100 | 78.2 | 21.5 (4/1) | 0.3 |

Relative percentage was calculated by dividing the area of each corresponding oligosaccharide peak by the total area of all peaks (×100) in the UPLC chromatograms. The reactions were performed using purified Trx-hAlg2 (5 ng/μL) incubated with the indicated concentrations of $M_1Gn_2$-PPhy substrate for 30 min. NA, not available.

concentration of 2 mM GDP-Man. As shown in Fig. 6c, d, the $K_m$ values for each of these two intermediates were calculated as 15.1 and 136.7 μM, respectively, suggesting that $M_2Gn_2$ (α-1,3) has a stronger affinity for hAlg2 than $M_2Gn_2$ (α-1,6) (~ 9 fold). On the other hand, to our surprise, the $k_{cat}$ value of $M_2Gn_2$ (α-1,6) (63.2 $min^{-1}$) is higher than that of $M_2Gn_2$ (α-1,3) (5.4 $min^{-1}$), resulting in a comparable or even higher $k_{cat}/K_m$ value (≈0.462) than that of $M_2Gn_2$ (α-1,3) (≈0.358). Generally, a higher $k_{cat}/K_m$ value reflects a better catalytic efficiency of substrate conversion by enzyme in a reaction. Apparently, this generality does not apply to the in vivo situation of hAlg2 because it catalyzes four MTase reactions in two steps of conversion ($M_1Gn_2$ to $M_2Gn_2$ and $M_2Gn_2$ to $M_3Gn_2$), in which $M_2Gn_2$ (α-1,6) only works as a minor substrate for the second conversion (Fig. 5a). Due to its low concentration, the minor intermediate $M_2Gn_2$ (α-1,6) could barely reach the maximum reaction velocity, although it possesses an even better catalytic efficiency than that of $M_2Gn_2$ (α-1,3). Nevertheless, our results demonstrated markedly different binding affinities of hAlg2 for each of its secondary substrates under these assay conditions and provide the explanation for why the $M_2Gn_2$ (α-1,3) → $M_3Gn_2$ biosynthetic route is the preferred route. Since $M_2Gn_2$ (α-1,3) binds to hAlg2 immediately after it is produced, the next α1,6-Man can be transferred onto it efficiently. In contrast because $M_2Gn_2$ (α-1,6) binds hAlg2 poorly, it is an inefficient competitor and thus remains unreactive.

**Excessive GDP-Man and high $M_1Gn_2$-PPhy levels increase the production of $M_2Gn_2$ (α-1,6) intermediate**. To gain insight into the basis for the preferential bias of hAlg2 toward the addition of (α-1,3)- versus (α-1,6)-linked Man, we varied parameters predicted to fluctuate in vivo. The first of these was the concentration of donor GDP-Man substrate. The concentration of GDP-Man used in our standard Alg2 assay was 2 mM. This concentration is probably much higher than steady-state levels of GDP-Man in most eukaryotic cells, which appears to be in the μM range; from 8 μM in *Trypanosoma brucei* to 55 μM in *Leishmania major*[35], and 20 μM in mouse embryonic fibroblasts cells[36]. To explore the effect of GDP-Man, hAlg2 activities were measured as a function of GDP-Man concentrations. As shown in Table 3, hAlg2 (5 ng/μL) was incubated with the substrate $M_1Gn_2$-PPhy (20 μM) for 30 min and various lower concentrations of GDP-Man. Preliminary work used a time course to measure the reaction time with varied hAlg2 concentrations (Supplementary Fig. 4). These results indicated 30 min as the most suitable time point for studying the effects of substrate/donor concentrations because within 30 min, $M_2Gn_2$ intermediates accumulated with relatively minor levels of $M_3Gn_2$ product formation. After the incubation, the ratio of $M_1Gn_2$ substrate, the $M_2Gn_2$ intermediates and the $M_3Gn_2$ product that remained in these reactions was then measured by UPLC-MS. At low concentrations of GDP-Man (5, 10,

and 25 μM), the reaction generated $M_2Gn_2$ (α-1,3) and trace amount of $M_3Gn_2$, while no $M_2Gn_2$ (α-1,6) was detected. Higher levels of GDP-Man (200 and 2000 μM) accelerated the reaction to produce $M_2Gn_2$ (α-1,3) and $M_3Gn_2$. These results indicate that hAlg2 can produce $M_3Gn_2$ through the $M_2Gn_2$ (α-1,3) intermediate (Fig. 5a; step 1 → 2) under a broad range of GDP-Man concentrations. On the other hand, $M_2Gn_2$ (α-1,6) formation was not observed even when the GDP-Man was raised up to 200 μM. When GDP-Man concentration was raised to 2000 μM (the standard concentration used in our in vitro assay), $M_2Gn_2$ (α-1,6) could be detected (5.6 %), demonstrating that the $M_2Gn_2$ (α-1,6) can be produced when GDP-Man is in excess.

To further test if $M_2Gn_2$ (α-1,6) production is influenced by LLO acceptor concentration, hAlg2 (5 ng/μL) was incubated with different concentrations of $M_1Gn_2$-PPhy (20, 40 and 100 μM) and 25 μM GDP-Man (to mimic physiological levels). After 30 min, the ratio of $M_1Gn_2$ substrate, the $M_2Gn_2$ intermediates, and the $M_3Gn_2$ product were measured. As shown in Table 4, $M_2Gn_2$ (α-1,3) was still the major intermediate produced, while $M_2Gn_2$ (α-1,6) was barely observed at a low $M_1Gn_2$-PPhy concentration (20 μM). However, the proportion of $M_2Gn_2$ (α-1,6) increased to 2.6% at $M_1Gn_2$-PPhy concentration of 40 μM (α-1,3/α-1,6 = 38/1, in Table 4) and to 20% at 100 μM (α-1,3/α-1,6 = 4/1, in Table 4), indicating the amount of $M_2Gn_2$ (α-1,6) formed as an intermediate was dependent on LLO concentrations. In conclusion, our results demonstrated that the accumulation of $M_2Gn_2$ (α-1,6) is dependent on both $M_1Gn_2$-PPhy and GDP-Man concentrations.

## Discussion

Alg2 MTase catalyzes a pair of consecutive reactions, adding both α1,3- and α1,6-Man residues onto the $M_1Gn_2$-PDol to form the branched core Man(α1,3)Man(α1,6)-$M_1Gn_2$ ($M_3Gn_2$) structure in the early steps of the LLO biosynthesis. Among the various Alg GTases that contribute to LLO assembly, Alg2 is unique not only because it mannosylates two different linkages but also because it can do so in different order[19]. To explore the biological relevance and molecular basis of these alternative $M_3Gn_2$ biosynthetic routes during N-linked glycosylation, we developed a method that enabled a kinetic analysis of these reactions. This method was sufficiently sensitive to quantitate each of the separate intermediates and was made possible by the intrinsically enhanced stability of human Alg2 as compared to yeast Alg2. The discovery of this characteristic arose through our topological analyses of hAlg2.

Despite the evolutionary conservation of their dual enzymatic activities, we found that yeast and human Alg2 proteins display completely different membrane topologies. hAlg2 protein interacts with the ER membrane via a single amphiphilic α helix in a nonperipheral manner, with both its N- and C-termini facing the cytoplasm (Fig. 1d, Model 2). Sequence comparisons of Alg2 orthologues from a variety of plants and animals suggest the

single N-terminal hydrophobic domain seen in hAlg2 is the rule rather than the exception. Indeed, multiple hydrophobic domains appear restricted to fungal and certain protozoan Alg2. Although hAlg2 is far less hydrophobic than ScAlg2, it nevertheless is localized to the ER membrane (Figs. 1–3). Alg2 interacts with Alg1 and Alg11 as part of a large multimeric MTase complex[7], so it was of interest to compare the predicted topologies of yeast and human Alg1 and Alg11 orthologues (Supplementary Fig. 5). While Alg1 orthologues look similar to one another, yeast and human Alg11 differ markedly in their predicted topologies but in a manner reversed from Alg2; ScAlg11 has a single hydrophobic region, while hAlg11 has four (Supplementary Fig. 5). This implies that on balance, both the human and yeast Alg1/2/11 MTase complex have a net similarity in the overall number of membrane-attaching regions, but the distribution of these domains evolved differently in each member of the complex. Further structural studies of the Alg1/2/11 multimeric complex in the context of the membrane may provide deeper insights into how these structural differences have allowed for the functional conservation of these orthologues. Nevertheless, the single membrane-attaching topology of hAlg2 facilitated efficient expression of hAlg2 in *E. coli* with high specific activity and protein stability.

Another critical aspect for the success of this in vitro Alg2 MTase assay was the repurposing of MGAT1 as a tool for separating the $M_2Gn_2$ (α1,3) and $M_2Gn_2$ (α1,6) intermediates in the reaction. MGAT1 converts high-mannose oligosaccharides to complex and hybrid structures on mature glycoproteins in the Golgi[31–33]. We showed that in vitro, MGAT1 efficiently added an α1,2-GlcNAc to both $M_2Gn_2$ (α1,3)-PPhy and $M_3Gn_2$-PPhy, suggesting MGAT1 substrate specificity requires only the presence of an α1,3-Man residue in the core structure of N-glycans (Fig. 5b, bottom panel). Exploiting this substrate specificity enabled the quantitative "tracking" of both $M_2Gn_2$ intermediates in time- or dose (enzyme)-dependent reactions by their unique molecular weight shifts as observed by UPLC-MS analysis (Fig. 6a, b). Reaction velocities determined from these experiments demonstrated that $M_2Gn_2$ (α1,3) is the major product of Alg2 MTase (Fig. 6a, b). Once made, $M_2Gn_2$ (α1,3) possesses a stronger affinity ($K_m$) to hAlg2 than $M_2Gn_2$ (α1,6) (Fig. 6c, d).

We hypothesize that this difference in $K_m$ explains why $M_2Gn_2$ (α1,3), but not $M_2Gn_2$ (α1,6) is preferentially converted to $M_3Gn_2$ in vivo, despite the observation that the calculated $k_{cat}/K_m$ values for both substrate are comparable or even better for the latter (Fig. 6c, d). First, both Alg2 MTase activities reside in a single catalytic site[19]. Thus, both $M_2Gn_2$ (α-1,6) and $M_2Gn_2$ (α-1,3) substrates must compete for the same active site in vivo. The comparable or better catalytic efficiency for $M_2Gn_2$ (α-1,6) is only possible in vitro when it was the only substrate present and at the same concentration as $M_2Gn_2$ (α-1,3) (as in Fig. 6c, d). However, in vivo, we believe that $M_2Gn_2$ (α-1,3), with a ~ 9 fold lower $K_m$, will outcompete $M_2Gn_2$ (α-1,6) for binding to the active site. Second, Alg2 MTase consecutively catalyzes $M_1Gn_2$ to $M_2Gn_2$ and $M_2Gn_2$ to $M_3Gn_2$ in vivo, so $M_2Gn_2$ (α-1,6) and/or $M_2Gn_2$ (α-1,3) substrates would exist as intermediates. However, no detectable $M_2Gn_2$ (α-1,6) could be observed under physiological concentrations of GDP-Man (i.e., low) (Table 3). If this is also true in vivo, then the $M_1Gn_2$ substrate would first form the enzyme substrate complex with hAlg2 and then be preferentially converted to $M_2Gn_2$ (α-1,3). Given its $K_m$, it seems reasonable to assume that this $M_2Gn_2$ (α-1,3) intermediate would remain in the enzyme substrate complex with hAlg2, and converted to the $M_3Gn_2$ final product, while $M_2Gn_2$ (α-1,6), because of its very low concentration and weaker affinity to hAlg2, cannot compete in the $M_3Gn_2$ biosynthetic route. Thus, our kinetic data point to the $M_1Gn_2 \rightarrow M_2Gn_2$ (α1,3) $\rightarrow M_3Gn_2$ reaction order (step 1 → 2 route in Fig. 5a) as the major route for hAlg2 MTase. Finally, the in vivo

MTase activities are likely to be much more complicated than those observed in vitro. In vivo, Alg2 works as multimeric MTase complex with Alg1 and Alg11 proteins. It seems likely that cooperativity between members of this multimeric Alg1/2/11 enzyme complex may also contribute to the molecular regulation of these reactions. Cryogenic electron microscopy or other structural analyses of Alg1/2/11 complex may shed light on these ideas.

Despite the drastic structural divergence of Alg2 orthologues, the minor $M_1Gn_2 \rightarrow M_2Gn_2$ (α1,6) Alg2 activity (step 3 → 4 route in Fig. 5a) has been evolutionarily conserved and therefore is likely to be of fundamental importance. Although the biological role of this minor Alg2-mediated route remains unknown, we speculate that it occurs in vivo, but only under abnormal conditions or at an undetectable level. One possibility is that this minor route provides a way for cells to regulate flux through the LLO biosynthetic pathway or that accumulation of $M_2Gn_2$ (α1,6) serves as warning signal. Based on our in vitro results, the reaction parameters that strongly triggered $M_2Gn_2$ (α1,6) production were GDP-Man and $M_1Gn_2$-PPhy (Tables 3 and 4) raising the interesting possibility that excessive GDP-Man and $M_1Gn_2$-PDol levels in vivo may similarly boost $M_2Gn_2$ (α1,6) production. Thus far, the only documented in vivo existence of the Alg2 minor route, whose hallmark is the accumulation of $M_2Gn_2$ (α1,6)-PDol, is in the temperature-sensitive yeast *alg2-1* mutant under non-permissive conditions[37]. Consistent with our in vitro kinetic results (Table 4), $M_1Gn_2$-PDol is also highly elevated in *alg2-1* yeast. There are so far no published reports of $M_2Gn_2$ (α1,6)-PDol detection in normal mamalian cells or tissues. However, $M_1Gn_2$-PDol and $M_2Gn_2$-PDol accumulation has been observed in skin fibroblasts of an ALG2-CDG patient[13]. This phenotype is reminiscent of the yeast *alg2-1* mutant[11,12,37]. While there is no direct evidence that the $M_2Gn_2$-PDol that accumulates in ALG2-CDG is enriched in $M_2Gn_2$ (α1,6), based on our kinetic analysis demonstrating the slow conversion of this intermediate to $M_3Gn_2$ product, we predict that this is likely the case. The kinetic system we described provides the tool to test this prediction. In addition, it will also be of interest to apply this system to study the physiological and pathological roles of human Alg2 at the early steps of LLO biosynthesis. Nevertheless, our results support a model in which Alg2 acts as a bifunctional sensor to regulate flux through the LLO biosynthetic pathway by reliance on alternative routes to control accumulation of the limiting $M_2Gn_2$ (α-1,6).

## Methods

**Plasmids, strains, cell line and culture conditions**. Plasmids used in this work and their important features are listed in Table 1. Standard molecular biology techniques were used for all plasmid constructions. Sequences of primers used in construction are available upon request.

*Saccharomyces cerevisiae* W303a (*MATa his3-11 leu2-3, 112 ura3-1 trp1-1 ade2-1 can1-100*) was used as the parental strain in this study. W303a-*GALpr-ALG2* contains a replacement of the chromosomal *ALG2* promoter with the glucose-repressible *GAL1/10* promoter and was used to test the activity of hAlg2 variant proteins by complementation of the glucose-dependent loss of *ALG2*. Yeast cells were cultured the YPA medium (1% yeast extract, 2% peptone, 50 mg/L adenine) supplemented with 2% glucose (YPAD) or 2% galactose (YPAG). Standard yeast growth conditions and genetic techniques were used.

HEK293 cells were cultured in Dulbecco's Modified Eagle medium (DMEM) containing 10% (vol/vol) fetal bovine serum (FCS, Biological Industries). G418 (600 µg/mL), puromycin (1 µg/mL), and streptomycin/penicillin (1 µg/mL) were used when necessary. Cells were maintained at 37°C under a humidified atmosphere with 5% $CO_2$. Transfection of DNA constructs was performed using Lipofectamine 2000 (Invitrogen) according to manufacturer's instruction.

**Immunofluoresecnce**. To detect the subcellular localization of FLAG-tagged hAlg2 and or mutant hAlg2 variants, HEK293 cells were transfected with plasmids expressing these proteins. 36 h after transfection, cells were harvested and re-plated onto glass coverslips pretreated with 1% gelatin and cultured 37°C for 1 day. The cells were fixed in 4% paraformaldehyde for 20 min, washed with PBS, and incubated with 40 mM ammonium chloride for 10 min. Subsequently, cells were permeabilized with 0.2% saponin diluted in PBS containing 5% FBS (blocking buffer)

for 5 min at room temperature. Permeabilized cells were further blocked with 1% BSA, 0.1% NaN₃ in PBS for 30 min. Calnexin was used as an ER marker. Mouse monoclonal anti-FLAG (M₂; F3165; Sigma-Aldrich) and rabbit polyclonal anti-calnexin (C4731; Sigma) were used as the primary antibodies. These antibodies were diluted 300-fold in blocking buffer and incubated with permeabilized cells for 1 h. After gently washing with PBS twice, cells were incubated for 1 h with 1000-fold-diluted fluorochrome-coupled secondary antibodies (Alexa Fluor 488-conjugated goat anti-mouse IgG; A48286 and Alexa Fluor 555-conjugated goat anti-rabbit IgG; A48283, Invitrogen). The coverslips were mounted on slides using ProLong Gold antifade Mountant with DAPI. Stained cells were visualized using a confocal microscope (C2si; NIS-Elements AR 4.30.00; Nikon) with a CFI Plan Apochromat VC oil objective lens (100× magnification and 1.4 NA) or FV1000 (Olympus) with a UPLSAPO oil lens (100× magnification and 1.4 NA).

**Preparation and analyses of membrane extracts from HEK293 and yeast cells**. HEK293 cells cultured in 6-well dishes were harvested and washed in PBS. The cell pellet was solubilized by resuspension in HEPES buffer with detergent [25 mM HEPES (pH 7.4), 150 mM NaCl, 1% NP-40, 1 mM PMSF and 1×Protein inhibitor cocktail] and incubation on ice for 30 min. Cell lysates were centrifuged at 15,000 $g$ for 10 min at 4 °C. The resulting supernatants were considered whole cell protein extracts. Subcellular fractionation of membrane fractions was performed as follow. HEK293 cells were homogenized in buffer lacking detergent [10 mM HEPES-KOH (pH 7.5), 0.22 M mannitol, 0.07 M sucrose] and lysates were centrifuged (3,000 $g$, 10 min, 4 °C). The resulting supernatants were subjected to ultracentrifugation (100,000 × $g$, 1 h, 4 °C), giving a pellet enriched for ER proteins (M) and supernatant with soluble proteins (S). For analysis of ER membrane proteins, the M fraction was either resuspended in 100 μL of TE (pH 7.2) containing 0.2% SDS or in 0.1 M Na₂CO₃ (pH 11.0), incubated on ice for 30 min, and further centrifuged at 100,000 × $g$ for 15 min at 4 °C. The supernatants were collected and the pellets resuspended with the same volume (100 μL) of TE buffer.

Proteins were extracted from yeast as previously described[38,39]. Briefly, cells were lysed in lysis buffer [0.2 M sorbitol, 1 mM EDTA, 50 mM Tris–HCl (pH 7.5), 1 mM PMSF and 1×Protease inhibitor cocktail] by vortexing with glass beads (425–600 mm; Sigma-Aldrich). The cell lysates were centrifuged at 3,000 $g$ for 10 min to remove cell debris. The ER-enriched fractions were obtained by further centrifugation (13,000 $g$, 30 min, 4 °C) of the resulting supernatant.

For western blotting, proteins were subjected to 10% SDS-PAGE and transferred to PVDF membranes (Bio-Rad). Mouse anti-FLAG (Sigma) and Rabbit anti-calnexin (CANX) (C4731; Sigma) antibodies were used at 1:2000 dilutions. Goat anti-mouse and goat anti-rabbit IgG-HRP (HS201-01 and HS101-01; Transgen) were used as secondary antibodies at 1:3000 dilution. Signals were visualized by Clarity Western ECL Substrate (Bio-Rad) and images were obtained by using 5200Multi (Tanon).

**Fluorescence protease protection (FPP) assay**. The FPP method refers to an assay previously described by Lorenz et al.[25,26]. HEK293 cells transiently transfected with plasmids were cultured to 60–70% confluence on coverslips onto glass coverslips which were put on the bottom of multi-well dishes. After removal of culture medium, cells were washed three times for 1 min each in KHM buffer (110 mM potassium acetate, 20 mM HEPES, 2 mM MgCl₂) at room temperature. Cells were kept in KHM buffer and directly imaged microscopically in a "pre-permeabilization" state. To permeabilize the plasma membrane, KHM buffer containing 20 μM digitonin was added and images were recorded as the "post-permeabilization" state. At this point, cells were washed with KHM buffer three times and treated with 50 μg/mL proteinase K in KHM buffer. Coverslips were removed and cells imaged using confocal laser scanning microscopy. For each condition, multiple coverslips were imaged (≥30 cells per coverslip) under identical settings (NIS-Elements Viewer 4.20; Nikon). Fluorescence intensity was determined using SimplePCI™ software (Hamamatsu).

**Expression and purification of recombinant proteins in E. coli**. All Alg2 proteins used in this study expressed in E. coli contain an N-terminal His6-Trx tag. A series of E. coli pET32 expression plasmids, containing ScAlg2, ScAlg2G257P, hAlg2, ScAlg1ΔTM, and hMGAT1ΔTM genes were transformed into E. coli Rosetta (DE3) cells (Merck) and cultured for expression of recombinant proteins. Recombinant proteins were extracted and purified with a HisTrap HP affinity column (GE Healthcare Life Sciences), as described previously[19]. Purified proteins were analyzed by SDS-PAGE, followed by staining with Coomassie Brilliant Blue R-250. Protein concentration was determined by the bicinchoninic acid assay (Sangon Biotech).

**In vitro MTase assay of hAlg2**. hAlg2 expressed and purified from E.coli was assayed in vitro as previously reported for ScAlg2[19]. Briefly, the substrate M₁Gn₂-PPhy was synthesized from Gn₂-PPhy by mannosylation with yeast Alg1 as described[30]. The standard reaction mixture in a final volume of 0.05 mL contained the following: 14 mM MES/NaOH (pH 6.0), 4 mM potassium citrate, 10 mM MgCl₂, 0.05% NP-40, 100 μM M₁Gn₂-PPhy, 2 mM GDP-Man, 1 M sucrose. After

adding various amounts of recombinant Alg2 protein, the reaction was carried out at 30 °C or 37 °C for various time periods and terminated by heating at 100 °C for 2 min. These standard reaction conditions were varied with different parameters including substrate concentration, enzyme concentration, and reaction time.

Reaction rates were calculated by measuring the ratio of substrate, intermediates and product in each reaction, in which either time or enzyme concentration was varied. Timed reactions contained 100 μM M₁Gn₂-PPhy and 1 ng/μL of purified Trx-hAlg2, and were terminated after 10, 20, 30, and 60 min. Enzyme varied reactions contained 100 μM M₁Gn₂-PPhy for 30 min, catalyzed by 1, 10, or 100 ng/μL of purified Trx-hAlg2. The two M₂Gn₂ intermediates, i.e., Man-(α1,3)Man-(β1,4)-Gn-(β1,4)-Gn-PP-phytanyl [M₂Gn₂(α1,3)-PPhy] and Man-(α1,6)Man-(β1,4)-Gn-(β1,4)-Gn-PP-phytanyl [M₂Gn₂(α1,6)-PPhy] were distinguished from one another by the selective specificity conferred by MGAT1, which recognizes M₂Gn₂(α1,3)-PPhy as a substrate for GlcNAc attachment but not M₂Gn₂(α1,6)-PPhy. hMGAT1ΔTM (200 ng/μL) was added to the quenched Trx-hAlg2 reactions that were further incubated with 1 mM UDP-GlcNAc and 10 mM MnCl₂ at 37 °C for 6 h to allow complete GlcNAc transfer to all a1,3 mannoses.

To analyze the kinetics of the second steps in Trx-hAlg2 reaction, the two intermediates M₂Gn₂(α1,3)- and M₂Gn₂(α1,6)-PPhy were prepared as follows: M₂Gn₂(α1,3)-PPhy was synthesized from Gn₂-PPhy by using ScAlg1ΔTM and Trx-ScAlg2G257P, a mutant we previously identified as only producing the M₂Gn₂(α1,3) intermediate;[19] M₂Gn₂(α1,6)-PPhy was synthesized from Gn₂-PPhy by using ScAlg1ΔTM, Trx-hAlg2 and α1,2-3-Mannosidase (New England Biolabs) as described below (see **PNGase F and Mannosidase treatments**). The intermediates were used for kinetic analyses without further purification. The apparent $K_m$ and $k_{cat}$ values for M₂Gn₂(α1,3)-PPhy (3.2–12 μM) and M₂Gn₂(α1,6)-PPhy (8–200 μM) were calculated by nonlinear regression curve fitting (GraphPad Prism 7.0, GraphPad Software Inc.), with 5 ng/μL of Trx-hAlg2 and reaction time of 10 min at 37 °C.

To analyze the effect of GDP-Man, reactions were performed with 20 μM M₁Gn₂-PPhy, 5 ng/μL of Trx-hAlg2 and GDP-Man concentrations that ranged from 25 μM to 2 mM, and were terminated after 30 min. The ratio of the two intermediates were calculated using the method described above. The ratio of the substrate, intermediates or the product was also measured in reactions with M₁Gn₂-PPhy (20, 40, or 100 μM) contained GDP-Man (25 μM) and 5 ng/μL of purified Trx-hAlg2.

**UPLC-MS analysis of oligosaccharides**. To release glycans from P-Phy, an equal volume of 20 mM HCl was added to each reaction and incubated at 100 °C for 1 h. The water-soluble fraction was desalted by solid-phase extraction using 1 mL Supelclean ENVI-Carb slurry (Sigma) equilibrated with 2% acetonitrile. The column was washed with 2% acetonitrile (10 mL), and oligosaccharides were eluted with 25% acetonitrile (3.0 mL) and lyophilized. Desalted oligosaccharides were injected into a Dionex Ultimate 3000 UPLC (Thermo Scientific; conditions: column, Waters Acquity UPLC BEH Amide Column 1.7 μm 2.1 × 100 mm; eluent A, CH₃CN; eluent B, H₂O; gradient: 0–2 min, 20% B; 2–15 min, 20–50% B; 15–18 min, 50% B; flow rate, 0.2 mL/min). The ESI-MS of eluate was measured on a TSQ Quantum Ultra (Thermo Scientific) in the mass range of 400–1600 (m/z, positive mode). Oligosaccharide transfer rate was quantified by calculating the peak intensity in LC-ESI-MS using Xcalibur (Version 2.2, Thermo Scientific).

**PNGase F and mannosidase treatments**. In total 10 μL of total reaction mixture contained 7.5 μL of sample, 1 μL of 10× Glyco Buffer2, 1 μL of 10% NP-40, and 0.5 μL of PNGase F (New England Biolabs). Reactions were incubated at 37 °C for 1 h. Digestion of glycans (100 nM) with 6.4 U of α1,2-3-mannosidase (New England Biolabs) and 4 U of α1,6-mannosidase (New England Biolabs) was performed at 25 °C for 16 h in 10 μL (total reaction volumes) with buffers supplied by the manufacturer.

**Statistics and reproducibility**. Kinetic analysis experiments were independently repeated three times. Sample size, mean, and standard deviation (SD) value are provided in the figure and figure legend. Kinetic parameters were determined by nonlinear regression fitted to Michaelis–Menten equation in GraphPad Prism 7.

**Reporting summary**. Further information on research design is available in the Nature Research Reporting Summary linked to this article.

## Data availability

Uncropped blots and gels are provided as Supplementary Fig. 6 and Supplementary Fig. 7 in Supplementary Information. The source data for the graphs and charts in the figures are available in Supplementary Data 1. All remaining data that support the results of this study are available from the corresponding authors upon reasonable request.

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

## Acknowledgements

We thank Drs. Hideki Nakanishi and Morihisa Fujita for discussion. We are grateful to Ms. Haili Liu for the arrangement of the data collection facilities. This work was supported by grants-in-aid from the National Natural Science Foundation of China (21778023; 21807048; 31971216; 22077053), Jiangsu Planned Projects for Postdoctoral Research Funds (2020Z167), Collaborative Innovation Center of Jiangsu Modern Industrial Fermentation, Top-notch Academic Programs Project of Jiangsu Higher Education Institutions, Program of Introducing Talents of Discipline to Universities (No. 111-2-06), Fundamental Research Funds for the Central Universities (JUSRP221011), Shandong Provincial Major Scientific and Technological Innovation Project (2019JZZY011006), Special fund for Zaozhuang Excellence agglomeration project to X-D Gao and Qing Lan Project of Jiangsu Province to N. Wang.

## Author contributions

M.-H.X., X.-X.X., C.-D.W., S.C., S.X., X.-Y.X., and N.W. performed experiments and analyzed data. N.W., N.D., and X.-D.G. proposed and supervised the project and wrote the manuscript. All authors confirmed and edited the manuscript.

## Competing interests

The authors declare no competing interests.
