## [Peer Review File · Communications Biology]

Reviewers' comments:

Reviewer #1 (Remarks to the Author):

This is a rigorous, important study on a timely topic. The authors used multiple valid methods and careful controls. A pleasure to review. Well written, with carefully prepared figures and tables.

They analyzed 2 models for human Alg2 anchoring (TM domain vs. membrane binding domain) that differs from yeast. They convincingly implicated a cytosolic membrane binding domain (pictured schematically as an amphipathic helix).

Then multiple substrates and intermediates were assessed to measure kinetic parameters, with a clever retooling of MGAT1 to distinguish M2 isomers. The effect of GDP-mannose concentration on 1,3 vs. 1,6 selectivity was very convincing, and suggest that in vivo 1,3 is almost always added first, and 1,6 added second.

Conclusions were also drawn with varying "C20" M1Gn2-PPhy acceptor concentrations. However, since this is not the natural "C95" M1Gn2-PP-Dol acceptor, and the natural dolichol-type acceptor likely has very different membrane interaction properties, it may be premature to draw too many conclusions about the role of the physiological acceptor concentration.

The summarizing statement clears up a mystery in the field, and has valuable impact for this pathway in general and the interpretation of Alg2-CDG genetic disorders: "Since M2Gn2 (α -1,3) binds to hAlg2 immediately after it is produced, the next α 1,6-man can be transferred on to it efficiently. In contrast because M2Gn2 (α -1,6) binds hAlg2 poorly, it is an inefficient competitor and thus remains unreactive."

In summary, well done! My only suggestions involve editorial adjustments to the manuscript.

- 1) Do you think the human Alg2 structure generally applicable to all metazoans?
- 2) For UPLC-MS why are the glycan peaks all dimers? Perhaps due to two isomers that occur during labeling of reducing ends?
- 3) Please discuss the expected structure of the cytosolic MBD. From the schematic diagram it is displayed as an amphipathic helix, but could it be a uniformly hydrophobic "membrane dip"?
- 4) Please use subscript (i.e. small and near the bottom of the text line) numerals which are standard for glycan terminology, such as in M1Gn2. In Microsoft Word you can make these substitutions efficiently through the text and tables.
- 5) Please prove better primary references for the statement "In addition, Alg1, Alg2 and Alg11 physically interact with one another and work as an MTase complex 10,11." Ref 10 from the title does not appear relevant, and Ref 11 appears to be a review article". Was Ref 7 meant?
- 6) Please aid the reader with a little more information about Man-(β 1,4)-Gn2-PPhy directly in the Results. What is PPhy, and why is it a valid substitute for dolichol-P?

Reviewer #2 (Remarks to the Author):

Comments on "Topological and enzymatic analysis of human Alg2 mannosyltransferase" by Xiang et al.

This manuscript describes the characterization of human Alg2 protein (hAlg2). The authors showed that its structure is quite distinct from the well-studied yeast homologues, and convincingly showed that hAlg2 associates with the ER through a peripheral binding of hydrophobic domain to the ER membrane. This human enzyme turned out to be much more stable than the yeast

orthologue, and the activity was modulated by the concentration of donor or acceptor substrate. Most of experiments presented sound and I believe that this manuscript can become acceptable after the following points are properly revised/clarified.

Major criticisms:

1. I do not understand the logics about the kinetics shown in Figures 6C and D. It appears that the authors want to claim that, due to the high K_m against α 1,6-linked Man₂GlcNAc₂, this minor intermediate is very inefficiently converted to Man₃GlcNAc₂. Nevertheless, Figure 6C/D showed that actually the catalytic efficiency (k_{cat}/K_m) is even BETTER when α 1-6 linked Man₂GlcNAc₂ is the substrate, when compared with that for α 1-3 linked Man₂GlcNAc₂ (if you zoom into [S] 0-10 μ M range, the reaction efficiency between the two reactions should not be that different – or may be even better for α 1-6 linked Man₂GlcNAc₂: α 1-3 Mannosyltransferase IF k_{cat}/K_m values stand true).

I therefore have to wonder - Aren't there any more complicated situation need to be considered, such as cooperativity of multimeric enzyme etc ? I understand that the authors do not assume that is the case, but just based on the kinetical data presented, it does not make sense to me. Am I missing a point?

2. I am also not sure about the data presentation of Figures 7A/B; These data show the ratio of Man₂ (α 1-6) and Man₂ (α 1-3) at ONE time point under various different donor/acceptor concentrations. To make this comparison reasonable, I have to assume that under these conditions they are majoring initial velocity of both Man₂ formations and no Man₃ has been formed (otherwise there is no guarantee that the ratio they observe faithfully reflect the kinetic comparison between two transferase activities), but it does not appear to be the case (Table 3)).

Then I do not think this comparison is not so meaningful, isn't it? Instead, the authors should take time course to compare the relative velocity comparison on the formation of Man₂ (α 1-6) and Man₂ (α 1-3), in order to make the analysis some sense.

Minor points:

Page 7, line 166; --, while GFP was used as a cytoplasmic marker that should be susceptible to added protease. -> GFP was used as a cytoplasmic marker that should be released upon permeabilization (?). (I thought GFP will be released upon permeabilization so protease won't act on that).

Page 7, line 173; until 50s -> by 50s (?)

Page 8, line 194; see Materials and Methods for details -> see Materials and Methods for details (non italic)

Page 11, line 289; formation was not be observed -> formation as not observed

Page 14, line 365; reminiscent the yeast *alg2-1* mutant -> reminiscent of the yeast *alg2-1* mutant

Page 15, line 405; IgG (Invitrogen) -> IgG (Invitrogen)) (one more parentheses?).

Figure 1: Figure 1B and C have been mislabeled.

Figure 3; the authors use mRFP-KDEL as a control but it would have been better to use GFP-KDEL as a control, as the susceptibility towards protease may be distinct between mRFP and GFP (which I do not believe that is the case..).

Figure 5; I assume that the author is sure that 6 h is enough time to achieve complete conversion of α 1,3M₂G_n2 but at least that should have been experimentally evaluated. If the authors have done so, they should at least mention that in the text.

Reviewer #3 (Remarks to the Author):

Dear Sir/Madam,

In Topological and enzymatic analysis of human Alg2 mannosyltransferase Xiao-Dong Gao and co-workers study the reactions kinetics of human ALG2 and shed light on its topology.

The major claim of the paper is that human ALG2 (hALG2) exhibits a single membrane-binding domain and is more stable. The authors further conclude that excess GDP-Man determines the reactions kinetics of hALG2.

The work is novel and will be of interest to the scientific community. All conclusions are original.

However, the results are not very convincing and the evidence presented is rather weak.

One of the major conclusions regarding the reaction kinetics of hALG2 is derived from the developed assay. The assay depends on extensive sample preparation, yet no evidence is presented that the assay can indeed be used for reliable quantitative measurements. In fact, I have great doubts whether this is possible, as MS measurements are in general not suitable for quantitative measurements. The absence of any standard errors for the data impede the study further (line 203 ff).

Moreover, it is insufficiently explained to the reader why the invertase fused to ALG2 would be glycosylated depending on the orientation. Moreover, using SDS-gel is a very indirect proof for N-glycosylation. Why where the N-glycans not determined by MALDI-MS?

Reviewers' comments:

Reviewer #1 (Remarks to the Author):

This is a rigorous, important study on a timely topic. The authors used multiple valid methods and careful controls. A pleasure to review. Well written, with carefully prepared figures and tables.

They analyzed 2 models for human Alg2 anchoring (TM domain vs. membrane binding domain) that differs from yeast. They convincingly implicated a cytosolic membrane binding domain (pictured schematically as an amphipathic helix).

Then multiple substrates and intermediates were assessed to measure kinetic parameters, with a clever retooling of MGAT1 to distinguish M2 isomers. The effect of GDP-mannose concentration on 1,3 vs. 1,6 selectivity was very convincing, and suggest that in vivo 1,3 is almost always added first, and 1,6 added second.

Conclusions were also drawn with varying "C20" MIGn2-PPhy acceptor concentrations. However, since this is not the natural "C95" MIGn2-PP-Dol acceptor, and the natural dolichol-type acceptor likely has very different membrane interaction properties, it may be premature to draw too many conclusions about the role of the physiological acceptor concentration.

The summarizing statement clears up a mystery in the field, and has valuable impact for this pathway in general and the interpretation of Alg2-CDG genetic disorders: "Since M2Gn2 (α -1,3) binds to hAlg2 immediately after it is produced, the next α 1,6-man can be transferred on to it efficiently. In contrast because M2Gn2 (α -1,6) binds hAlg2 poorly, it is an inefficient competitor and thus remains unreactive."

In summary, well done! My only suggestions involve editorial adjustments to the manuscript.

We thank this reviewer for the positive and constructive comments for our study

1) Do you think the human Alg2 structure generally applicable to all metazoans?

Our response:

Yes, like human Alg2, all orthologous Alg2 proteins in animals and plants have only one hydrophobic domain at their N-terminal regions. Multiple hydrophobic domains are only found in Fungi and in some Protista. We tested some of those Alg2 proteins (more than 10 Alg2 homologs from different species), and confirmed their dual MTase activities (unpublished results).

We added the following sentence in the revised manuscript to reflect this point (Page 13; lines 336-339).

Sequence comparisons of Alg2 orthologues from a variety of plants and animals suggest the single N-terminal hydrophobic domain seen in hAlg2 is the rule rather than the exception. Indeed,

multiple hydrophobic domains appear restricted to fungal and certain protozoan Alg2.

2) *For UPLC-MS why are the glycan peaks all dimers? Perhaps due to two isomers that occur during labeling of reducing ends?*

Our response:

The glycans were separated by UPLC and instantly analyzed by ESI-MS to give the molecular weight information. That's true that the dual peaks were due to the α - and β -anomeric forms of released glycans.

We added the following sentence in the revised manuscript to reflect this point (Page 8; lines 100-201).

Noteworthy, dual peaks observed in UPLC-MS are likely the result of α - and β -anomeric forms of released glycans.

3) *Please discuss the expected structure of the cytosolic MBD. From the schematic diagram it is displayed as an amphipathic helix, but could it be a uniformly hydrophobic "membrane dip"?*

Our response:

Yes, as this reviewer indicated, helical wheel projection of the cytosolic MBD arranges almost all its hydrophobic residues to one side of the helix, suggesting an amphiphilic α helix structure. We have mutagenized total four of those hydrophobic residues to A (alanine), and confirmed that this substitution significantly affected the ER membrane localization of hAlg2p (data not shown). So, our results indicated that at least, the amphiphilic α helix of MBD is involved in its membrane interaction with ER. In addition, we couldn't confirm any other hydrophobic domain or residues in Alg2 that can affect membrane interaction with the MBD.

4) *Please use subscript (i.e. small and near the bottom of the text line) numerals which are standard for glycan terminology, such as in M1Gn2. In Microsoft Word you can make these substitutions efficiently through the text and tables.*

Our response:

We made these substitutions using subscript numerals.

5) *Please prove better primary references for the statement "In addition, Alg1, Alg2 and Alg11 physically interact with one another and work as an MTase complex 10,11." Ref 10 from the title does not appear relevant, and Ref 11 appears to be a review article". Was Ref 7 meant?*

Our response:

This is our mistake, and we have fixed it in the revised version. Ref 10 was replaced by Ref 7.

6) *Please aid the reader with a little more information about Man-(β 1,4)-Gn2-PPhy directly in the Results. What is PPhy, and why is it a valid substitute for dolichol-P?*

Our response:

We added the following sentences in the Results section (Pages 8, lines 191 to 198).

To assay MTase activities, purified recombinant proteins (40 ng/ μ L) were added to a standard reaction, in which Man-(β 1,4)-Gn₂-pyrophosphate-phytanyl (M₁Gn₂-PPhy) was used as the LLO acceptor instead of the natural M₁Gn₂-PDol acceptor (see *Materials and Methods* for details). Phytanol, with only four isoprene units, is much shorter than dolichol (C₈₅₋₁₀₅) and can be easily synthesised²⁸. Importantly, the first five MTases of lipid-linked oligosaccharide synthesis show no preference for dolichol- versus phytanol-linked substrates³. Therefore phytanol-linked oligosaccharides (PLOs) are widely used as acceptor substrates for enzymatic studies of Alg GTase^{3,19,29,30}.

Reviewer #2 (Remarks to the Author):

Comments on “Topological and enzymatic analysis of human Alg2 mannosyltransferase” by Xiang et al.

This manuscript describes the characterization of human Alg2 protein (hAlg2). The authors showed that its structure is quite distinct from the well-studied yeast homologues, and convincingly showed that hAlg2 associates with the ER through a peripheral binding of hydrophobic domain to the ER membrane. This human enzyme turned out to be much more stable than the yeast orthologue, and the activity was modulated by the concentration of donor or acceptor substrate. Most of experiments presented sound and I believe that this manuscript can become acceptable after the following points are properly revised/clarified.

Major criticisms:

1. *I do not understand the logics about the kinetics shown in Figures 6C and D. It appears that the authors want to claim that, due to the high Km against alpha1,6-linked Man2GlcNAc2, this minor intermediate is very inefficiently converted to Man3GlcNAc2. Nevertheless, Figure 6C/D showed that actually the catalytic efficiency (kcat/Km) is even BETTER when alpha1-6 linked Man2GlcNAc2 is the substrate, when compared with that for alpha1-3 linked Man2GlcNAc2 (if you zoom into [S] 0-10 microM range, the reaction efficiency between the two reactions should not be that different – or may be even better for alpha1-6 linked Man2GlcNAc2:alpha1-3 Mannosyltransferase IF kcat/Km values stand true).*

I therefore have to wonder - Aren't there any more complicated situation need to be considered, such as cooperativity of multimeric enzyme etc ? I understand that the authors do not assume that is the case, but just based on the kinetical data presented, it does not make sense to me. Am I missing a point?

Our response:

We thank the reviewer for pointing out that a better explanation of our interpretation of these kinetic results in Fig 6C and D is required. As the reviewer indicated, we interpreted the kinetic

data in Figure 6C and D, in particular the vastly different K_m values, as an explanation for why hAlg2 preferentially converts M_2Gn_2 (α -1,3) rather than M_2Gn_2 (α -1,6) to M_3Gn_2 *in vivo*, despite the observation that the calculated k_{cat}/K_m values for both substrate are comparable or even better for the latter { k_{cat}/K_m (≈ 0.462) for M_2Gn_2 (α -1,6) vs (≈ 0.358) for M_2Gn_2 (α -1,3)}. Our model for how our *in vitro* results relate to hAlg2 MTase reactions *in vivo* are as follows :

1. As we showed previously, Alg2 MTase activities reside in a single catalytic site {Li, S.-T. et al. Alternative routes for synthesis of N-linked glycans by Alg2 mannosyltransferase. The FASEB Journal 32, 2492-2506 (2018)}. Thus, both M_2Gn_2 (α -1,6) and M_2Gn_2 (α -1,3) substrates must compete for the same active site *in vivo*. While the apparent *in vitro* Alg2 catalytic efficiency is better when alpha1-6 linked $Man_2GlcNAc_2$ is the substrate, this is only possible *in vitro* when M_2Gn_2 (α -1,6) was present as the only substrate and at the same concentration with M_2Gn_2 (α -1,3) (as in Figure 6C and D). However *in vivo*, we believe that M_2Gn_2 (α -1,3), with a ~ 9 fold lower K_m , will outcompete M_2Gn_2 (α -1,6) for binding to the active site.
2. Alg2 MTase consecutively catalyzes M_1Gn_2 to M_2Gn_2 and M_2Gn_2 to M_3Gn_2 *in vivo*, so M_2Gn_2 (α -1,6) and/or M_2Gn_2 (α -1,3) substrates would exist as intermediates. We demonstrated that hAlg2 prefers to convert M_1Gn_2 to the M_2Gn_2 (α -1,3) intermediate (Figure 6A and B) and under physiological concentrations of GDP-man (i.e. low), no detectable M_2Gn_2 (α -1,6) is observed (see Table 3). If this is true *in vivo*, then the M_1Gn_2 substrate would first form the ES complex with hAlg2, and be preferentially converted to M_2Gn_2 (α -1,3). Given its K_m , it seems reasonable to assume that this M_2Gn_2 (α -1,3) intermediate would remain in the ES complex with hAlg2, and converted to the M_3Gn_2 final product, while M_2Gn_2 (α -1,6), because of its very low concentration and weaker affinity to hAlg2, cannot compete in this M_3Gn_2 biosynthetic route.
3. Alg2 works as multimeric MTase complex with Alg1 and Alg11 proteins *in vivo*. Thus, as suggested by the reviewer, it seems likely that cooperativity between members of the multimeric Alg1/2/11 enzyme complex may also contribute to the molecular regulation of these reactions. Cryogenic electron microscopy or other structural analyses of Alg1/2/11 complex may shed light on this idea.

Nevertheless, combining with other results, our kinetic studies suggested $M_1Gn_2 \rightarrow M_2Gn_2$ (α -1,3) $\rightarrow M_3Gn_2$ biosynthetic route is the preferred route of hAlg2. To improve the statement on this issue, we added the following sentences in the results section (Pages 10 and 11, lines 271-280), and clarified how our *in vitro* results can be extrapolated to what is happening *in vivo*. in the discussion section (Pages 14 and 15, lines 363-385).

Pages 10 and 11, lines 271-280:

On the other hand, to our surprise, the k_{cat} value of M_2Gn_2 (α -1,6) (63.2 min^{-1}) is higher than that of M_2Gn_2 (α -1,3) (5.4 min^{-1}), resulting in a comparable or even higher k_{cat}/K_m value (≈ 0.462) than that of M_2Gn_2 (α -1,3) (≈ 0.358). Generally, a higher k_{cat}/K_m value reflects a better catalytic

efficiency of substrate conversion by enzyme in a reaction. Apparently, this generality does not apply to the *in vivo* situation of hAlg2 because it catalyzes four MTase reactions in two steps of conversion (M_1Gn_2 to M_2Gn_2 and M_2Gn_2 to M_3Gn_2), in which M_2Gn_2 (α -1,6) only works as a minor substrate for the second conversion (Fig. 5A). Due to its low concentration, the minor intermediate M_2Gn_2 (α -1,6) could barely reach the maximum reaction velocity, although it possesses an even better catalytic efficiency than that of M_2Gn_2 (α -1,3).

Pages 14 and 15, lines 363-385:

We hypothesize that this difference in K_m explains why M_2Gn_2 (α 1,3), but not M_2Gn_2 (α 1,6) is preferentially converted to M_3Gn_2 *in vivo*, despite the observation that the calculated k_{cat}/K_m values for both substrate are comparable or even better for the latter (Fig. 6C, D). First, both Alg2 MTase activities reside in a single catalytic site 19. Thus, both M_2Gn_2 (α -1,6) and M_2Gn_2 (α -1,3) substrates must compete for the same active site *in vivo*. The comparable or better catalytic efficiency for M_2Gn_2 (α -1,6) is only possible *in vitro* when it was the only substrate present and at the same concentration as M_2Gn_2 (α -1,3) (as in Figure 6C and D). However *in vivo*, we believe that M_2Gn_2 (α -1,3), with a ~ 9 fold lower K_m , will outcompete M_2Gn_2 (α -1,6) for binding to the active site. Second, Alg2 MTase consecutively catalyzes M_1Gn_2 to M_2Gn_2 and M_2Gn_2 to M_3Gn_2 *in vivo*, so M_2Gn_2 (α -1,6) and/or M_2Gn_2 (α -1,3) substrates would exist as intermediates. However, no detectable M_2Gn_2 (α -1,6) could be observed under physiological concentrations of GDP-man (i.e. low) (Table 3). If this is also true *in vivo*, then the M_1Gn_2 substrate would first form the enzyme substrate complex with hAlg2 and then be preferentially converted to M_2Gn_2 (α -1,3). Given its K_m , it seems reasonable to assume that this M_2Gn_2 (α -1,3) intermediate would remain in the enzyme substrate complex with hAlg2, and converted to the M_3Gn_2 final product, while M_2Gn_2 (α -1,6), because of its very low concentration and weaker affinity to hAlg2, cannot compete in the M_3Gn_2 biosynthetic route. Thus, our kinetic data point to the $M_1Gn_2 \rightarrow M_2Gn_2$ (α 1,3) $\rightarrow M_3Gn_2$ reaction order (step 1 \rightarrow 2 route in Fig. 5A) as the major route for hAlg2 MTase. Finally, the *in vivo* MTase activities are likely to be much more complicated than those observed *in vitro*. *In vivo*, Alg2 works as multimeric MTase complex with Alg1 and Alg11 proteins. It seems likely that cooperativity between members of this multimeric Alg1/2/11 enzyme complex may also contribute to the molecular regulation of these reactions. Cryogenic electron microscopy or other structural analyses of Alg1/2/11 complex may shed light on these ideas.

2. *I am also not sure about the data presentation of Figures 7A/B; These data show the ratio of Man2 (alpha1-6) and Man2 (alpha1-3) at ONE time point under various different donor/acceptor concentrations. To make this comparison reasonable, I have to assume that under these conditions they are majoring initial velocity of both Man2 formations and no Man3 has been formed (otherwise there is no guarantee that the ratio they observe faithfully reflect the kinetic comparison between two transferase activities), but it does not appear to be the case (Table 3)).*

Then I do not think this comparison is not so meaningful, isn't it? Instead, the authors should take time course to compare the relative velocity comparison on the formation of Man2 (alpha1-6) and Man2 (alpha1-3), in order to make the analysis some sense.

Our response:

We thank the reviewer for this valuable suggestion. As the reviewer indicated, Figures 7A/B show the ratio of M_2Gn_2 (α -1,6) and M_2Gn_2 (α -1,3) at 30 min under various conditions. We have done the time courses with different hAlg2 concentrations in the preliminary work. These results now have been added into the revised manuscript as the supplementary Figure 4 (Fig. S4). We selected 30 min as the reaction time point with the following concerns:

1. The time course results (Fig. S4) demonstrated that within 30 min, intermediate M_2Gn_2 was accumulated whereas M_3Gn_2 was relatively minor;
2. Product M_3Gn_2 is inevitably produced, no matter with short time (Fig. 6A, Fig. S4) nor low substrate/donor concentrations (Table 3). However, it has been demonstrated that $M_1Gn_2 \rightarrow M_2Gn_2$ (α -1,3) $\rightarrow M_3Gn_2$ is the preferred route of hAlg2, indicating the small amount of M_3Gn_2 detected in Fig. 7A and B was mainly generated from M_2Gn_2 (α -1,3).

Therefore, we chose 30 min as the time point to compare the relative ratio of M_2Gn_2 (α -1,6) and M_2Gn_2 (α -1,3), in order to reveal the effect of substrate/donor concentrations on the M_2Gn_2 formation. To improve the statement on this issue, we mainly made several changes in the revised manuscript as follow:

1. Two curves of time course have been added as the supplementary Figure 4 in order to explain why we have chosen 30 min as the reaction time for study the effects of substrate/donor concentrations;
2. We replaced Fig. 7A by Table 4 to show the relative percentage of each oligosaccharide in hAlg2 reactions;
3. Since reviewer #1 also suggested that it may be premature to draw physiological conclusions using unusual conditions (Fig. 7B, 2000 μ M GDP-Man), Figure 7B was omitted in the revised manuscript.

We have added following sentences in the manuscript for explaining these changes.
(Pages 11 and 12, lines 296-300)

Preliminary work used a time course to measure the reaction time with varied hAlg2 concentrations. These results indicated 30 min as the most suitable time point for studying the effects of substrate/donor concentrations because within 30 min, M_2Gn_2 intermediates accumulated with relatively minor levels of M_3Gn_2 product formation (Supplementary Fig. 4).

(Pages 12, lines 311-320)

To further test if M_2Gn_2 (α -1,6) production is influenced by LLO acceptor concentration, hAlg2 (5 ng/ μ L) was incubated with different concentrations of M_1Gn_2 -PPhy (20, 40 and 100 μ M) and 25 μ M GDP-Man (to mimic physiological levels). After 30 min, the ratio of M_1Gn_2 substrate, the M_2Gn_2 intermediates, and the M_3Gn_2 product were measured. As shown in Table 4, M_2Gn_2 (α -1,3) was still the major intermediate produced, while M_2Gn_2 (α -1,6) was barely observed at a low M_1Gn_2 -PPhy concentration (20 μ M). However, the proportion of M_2Gn_2 (α -1,6) increased to 2.6% at M_1Gn_2 -PPhy concentration of 40 μ M (α -1,3/ α -1,6 = 38/1, in Table 4) and to 20% at 100 μ M (α -1,3/ α -1,6 = 4/1, in Table 4), indicating the amount of M_2Gn_2 (α -1,6) formed as an intermediate was dependent on LLO concentrations. In conclusion, our results demonstrated that the accumulation of M_2Gn_2 (α -1,6) is dependent on both M_1Gn_2 -PPhy and GDP-Man concentrations.

Minor points:

Page 7, line 166; --, while GFP was used as a cytoplasmic marker that should be susceptible to added protease. -> GFP was used as a cytoplasmic marker that should be released upon permeabilization (?). (I thought GFP will be released upon permeabilization so protease won't act on that).

Our response: We thank this reviewer for pointing this out. We corrected the description for GFP as the following:

Page 7, lines 166-167:

while GFP was used as a cytoplasmic marker that should be released from the cytosol upon permeabilization.

Page 7, line 173; until 50s -> by 50s (?)

Our response: It has been changed to **by 50 sec** (page 7, line 173). Thanks for pointing this out.

Page 8, line 194; see Materials and Methods for details -> see Materials and Methods for details (non italic)

Our response: It has been fixed

Page 11, line 289; formation was not be observed -> formation as not observed

We changed to **“formation was not observed”** (page 12, line 307)

Page 14, line 365; reminiscent the yeast alg2-1 mutant -> reminiscent of the yeast alg2-1 mutant

Our response: It has been corrected.

Page 15, line 405; IgG (Invitrogen) -> IgG (Invitrogen)) (one more parentheses?).

Our response: It has been corrected.

Figure 1: Figure 1B and C have been mislabeled.

Our response: We apologize for our careless mistake.

Figure 3; the authors use mRFP-KDEL as a control but it would have been better to use GFP-KDEL as a control, as the susceptibility towards protease may be distinct between mRFP and GFP (which I do not believe that is the case).

Our response: We agree with this reviewer's opinion. To match up with the GFP-hAlg2 fusion

protein, GFP-KDEL is the better control. We chose mRFP-KDEL for the ER marker because the GFP has already been used as a cytoplasmic marker.

Figure 5; I assume that the author is sure that 6 h is enough time to achieve complete conversion of alpha1,3M2Gn2 but at least that should have been experimentally evaluated. If the authors have done so, they should at least mention that in the text.

Our response: We thank this reviewer for this suggestion. In our previous study, we have confirmed that this recombinant MGAT1 can completely convert a peptide-modified M3Gn2 to Gn-M3Gn2 form in 2 h. Therefore, we chose 3 times longer incubation for converting our M2Gn2-PPhy and M3Gn2-PPhy substrates.

We have added following sentences in the text (page 9, line 231-234).

In our previous study, we confirmed that under similar reaction conditions, purified recombinant MGAT1 completely converted peptide-modified M₃Gn₂ to Gn-M₃Gn₂ in 2 h³⁴. Therefore, in our assay, a 6 h incubation was chosen for converting M₃Gn₂- or M₂Gn₂-PPhy intermediates.

Reviewer #3 (Remarks to the Author):

Dear Sir/Madam,

In Topological and enzymatic analysis of human Alg2 mannosyltransferase Xiao-Dong Gao and co-workers study the reactions kinetics of human ALG2 and shed light on its topology.

The major claim of the paper is that human ALG2 (hALG2) exhibits a single membrane-binding domain and is more stable. The authors further conclude that excess GDP-Man determines the reactions kinetics of hALG2.

The work is novel and will be of interest to the scientific community. All conclusions are original.

However, the results are not very convincing and the evidence presented is weak.

One of the major conclusions regarding the reaction kinetics of hALG2 is derived from the developed assay. The assay depends on extensive sample preparation, yet no evidence is presented that the assay can indeed be used for reliable quantitative measurements. In fact, I have great doubts whether this is possible, as MS measurements are in general not suitable for quantitative measurements. The absence of any standard errors for the data impede the study further (line 203 ff).

Our response:

The LC-MS based quantitative activity assay was developed by our group and has been widely applied to many Alg proteins (Ref 3, 19, 24, 30). Compared with traditional two-step procedures (removal of lipid from oligosaccharides and oligosaccharide labeling prior to HPLC

analysis), LC-MS measures product formation directly. In this assay, amide normal-phase liquid chromatography (LC) gives the quantitative information of each oligosaccharide component in the reaction and the tandem coupled ESI-MS analyzes the oligosaccharide structure. Therefore, we believe the LC-MS based assay is a useful tool to quantitate Alg glycosyltransferase enzymatic activity *in vitro*.

Moreover, it is insufficiently explained to the reader why the invertase fused to ALG2 would be glycosylated depending on the orientation. Moreover, using SDS-gel is a very indirect proof for N-glycosylation. Why where the N-glycans not determined by MALDI-MS?

Our response:

We thank this reviewer for the questions. For the first question, yeast invertase (Suc2A) is well established readout for assessing membrane orientation of peptides fused to it, because invertase contains three robust NX(S/T) glycosylation sites that are glycosylated only when in the ER lumen. Experimental rationale and references (Page 5, lines 129-131) are in the manuscript. To address the second question, while MALDI-MS is a direct method to assess the presence of N-glycans, so too is SDS-PAGE analysis of PNGase-treated glycoproteins. PNGase-dependent electrophoretic shifts are widely accepted as proof of N-glycosylation because of the exquisite specificity of this amidase for cleavage between the innermost GlcNAc and asparagine residues of high mannose, hybrid, and complex oligosaccharides from N-linked glycoproteins.

REVIEWERS' COMMENTS:

Reviewer #1 (Remarks to the Author):

Thank you for your erudite responses to my questions. The authors are to be commended on this fine work.

Reviewer #2 (Remarks to the Author):

I am quite satisfied with the authors' revision. Now this manuscript is ready for publication.